# Flow-based Variational Mutual Information: Fast and Flexible Approximations

**Caleb Dahlke**
Department of Mechanical Engineering
University of Michigan
Ann Arbor, MI, USA
`cdahlke@umich.edu`

**Jason Pacheco**
Department of Computer Science
University of Arizona
Tucson, AZ, USA
`pachecoj@arizona.edu`

## Abstract

Mutual Information (MI) is a fundamental measure of dependence between random variables, but its practical application is limited because it is difficult to calculate in many circumstances. Variational methods offer one approach by introducing an approximate distribution to create various bounds on MI, which in turn is an easier optimization problem to solve. In practice, the variational distribution chosen is often a Gaussian, which is convenient but lacks flexibility in modeling complicated distributions. In this paper, we introduce new classes of variational estimators based on Normalizing Flows that extend the previous Gaussian-based variational estimators. Our new estimators maintain many of the same theoretical guarantees while simultaneously enhancing the expressivity of the variational distribution. We experimentally verify that our new methods are effective on large MI problems where discriminative-based estimators, such as MINE and InfoNCE, are fundamentally limited. Furthermore, we compare against a diverse set of benchmarking tests to show that the flow-based estimators often perform as well, if not better, than the discriminative-based counterparts. Finally, we demonstrate how these estimators can be effectively utilized in the Bayesian Optimal Experimental Design setting for online sequential decision making.

## 1 Introduction

Mutual Information (MI), a fundamental measure of dependence from information theory, has found utility in a variety of fields such as Bayesian optimal experimental design (Lindley, 1956; Foster et al., 2019), representation learning (Alemi et al., 2016; Chen et al., 2016), and structure learning (Vinh et al., 2011). Despite having a simple interpretation, MI typically lacks a closed form solution making it a difficult measure to use in many practical settings. Straightforward sample-based estimates can be used to approximate MI, but these are often inefficient both in terms of computation and sample complexity. Moreover, these so-called nested Monte Carlo (NMC) estimators exhibit large finite sample bias that decays slowly (Zheng et al., 2018; Rainforth et al., 2018).

To resolve the computational issues associated with MI, many so-called discriminative based estimators have been proposed. These distribution-free approximations typically utilize the Donsker-Varadhan lower bound of MI (Donsker & Varadhan, 1975), with some examples including MINE (Belghazi et al., 2018) and NWJ (Nguyen et al., 2010). While these estimators have seen wide adoption in practice, McAllester & Stratos (2020) show that they require an exponential number of samples to estimate large MI values. This poor sample complexity presents a fundamental limitation in regimes where the achievable information is likely to be high.

Generative variational methods, referred to simply as variational methods for the remainder of the paper, directly approximate the intractable target distribution with a tractable surrogate have seen promise in practical settings (Barber & Agakov, 2004; Foster et al., 2019; 2020). These variational methods make MI estimation more scalable than NMC based approaches and are not subject to the same fundamental limitation of approximating large MI as discriminative based estimators. Yet computation of these estimators can be prohibitively costly, particularly in sequential decision making regimes (Foster et al., 2021; Ivanova et al., 2021). Moreover, the accuracy of variational estimators is

heavily dependent on the expressibility of the chosen surrogate distribution, which is often limited to a Gaussian approximation to control computation (Dahlke et al., 2023; Pacheco & Fisher III, 2019; Barber & Agakov, 2004).

We address the computational and accuracy aspects of variational MI estimators by introduction of two new estimators based on normalizing flows (Kobyzev et al., 2021). The proposed estimators flexibly adapt to complex target distributions and allow a straightforward tradeoff between accuracy and computational complexity. The first estimator, *Joint Variational Flow (JVF)*, builds on the *Joint Variational Gaussian (JVG)* estimator proposed in Dahlke et al. (2023). That previous work established that, for Gaussian approximating families, the estimator can be easily computed by moment-matching the target distribution. We relax the Gaussian assumption and show that the same efficient updates can be used in a flow-based estimator while simultaneously achieving more flexible density approximation.

Both JVG and JVF assume access to a joint distribution of the random quantities of interest; Gaussian in the first case and in the second a more flexible generative distribution given by reversing the normalizing flow. While this assumption of an underlying joint leads to efficient computation, it can restrict the MI approximation. The *Neural Variational Gaussian (NVG)* relaxes this assumption using a neural-network parameterized Gaussian distribution to approximate the posterior. We extend this estimator to *Neural Variational Flow (NVF)* to achieve a more expressible estimator that incorporates flow-based density estimates into the underlying approximation.

**Contributions** We summarize the main results of our paper as follows. We introduce 2 families of variational MI estimators (JVF and NVF) that incorporate normalizing flows to flexibly adapt to complex multimodal target distributions. We extend existing results to show that parameters of the underlying flow distribution can be efficiently computed via moment-matching operations. We provide a wide array of benchmarking experiments to highlight the strengths and weaknesses of each estimator as well as to compare to many of the state of the art MI estimators. Finally, we show that our variational estimators flexibly adapt to complex sequential decision making tasks with nonlinear non-Gaussian noise distributions.

## 2 PRELIMINARIES

Consider a joint distribution $p(X, Y)$ over latent variable $X$ and observation variable $Y$. Mutual information (MI) measures the dependence between these random quantities and is given by:

$$I(X, Y) = H_p(p(X)) - H_p(p(X|Y)) \tag{1}$$

where $H_p(p(X)) = \mathbb{E}\left[-\log p(X)\right]$ is the *marginal entropy* and $H_p(p(X|Y)) = E\left[-\log p(X|Y)\right]$ is the *conditional entropy*. The entropy expectations are taken with respect to the joint $p(X, Y)$. Despite its simplicity, MI typically lacks a closed-form solution. Furthermore, nested Monte Carlo estimators of MI have large finite sample bias that is slow to decay (Zheng et al., 2018; Rainforth et al., 2018) making direct sample-based estimates of MI practically infeasible in many settings. To address these issues, variational methods introduce an approximate distribution, $q$, to estimate the underlying true distribution, $p$ by minimizing Kullback-Leibler divergence: $\min_q \text{KL}(q \| p)$.

### 2.1 VARIATIONAL MUTUAL INFORMATION

The MI in Eqn. (1) requires knowledge of the marginal and posterior distributions. We consider an approach where both distributions are approximated, $p(X) \approx q_{\text{marg}}(X)$ and $p(X|Y) \approx q_{\text{post}}(X|Y)$, in the *marginal and posterior* approximation (Foster et al., 2019; Dahlke et al., 2023):

$$I_{\text{m}+p}(X, Y) := H_p(q_{\text{marg}}(X)) - H_p(q_{\text{post}}(X|Y)) \approx I(X, Y) \tag{2}$$

where $H_p(q(\cdot)) = \mathbb{E}_p[-\log q(\cdot)]$ is the cross-entropy. We can then minimize the following upper bound on absolute error (Foster et al., 2019):

**Lemma 2.1.** *For any model $p(X, Y)$ and distributions $q_{\text{marg}}(X)$, $q_{\text{post}}(X \mid Y)$, the following holds:*

$$|I_{m+p} - I| \leq \min_{q_{\text{marg}}} H_p(q_{\text{marg}}(X)) + \min_{q_{\text{post}}} H_p(q_{\text{post}}(X \mid Y)) + C$$

*where $C = -H_p(p(X)) - H_p(p(X \mid Y))$ does not depend on $q_{\text{marg}}$ or $q_{\text{post}}$. Further, the RHS is $0$ iff $q_{\text{marg}}(X) = p(X)$ and $q_{\text{post}}(X \mid Y) = p(X \mid Y)$ almost surely.*

This bound can be made arbitrarily tight by finding good variational approximations of both $p(X)$ and $p(X \mid Y)$ and has the added benefit of only assuming that $p(X, Y)$ can be sampled from. Thus, $I_{m+p}$ can be applied in many settings, such as those with implicit (e.g. simulation-based) distributions.

## 2.2 JOINT VARIATIONAL GAUSSIAN (JVG) ESTIMATOR

Dahlke et al. (2023) show that under a Gaussian approximating distribution the upper bound in Lemma 2.1 can be efficiently minimized by moment-matching operations.

**Theorem 2.2** (Moment Matching = Optimization). *Let $q(X, Y)$ be a joint Gaussian density:*

$$q(X, Y) = \mathcal{N} \left( \begin{bmatrix} X \\ Y \end{bmatrix} \middle| \mu := \begin{bmatrix} \mu_x \\ \mu_y \end{bmatrix}, \Sigma := \begin{bmatrix} \Sigma_{xx} & \Sigma_{xy} \\ \Sigma_{yx} & \Sigma_{yy} \end{bmatrix} \right) \tag{3}$$

*then the marginal and posterior optimizing the bound in Lemma 2.1 are given by:*

$$q_{\mathrm{marg}}(X) = \mathcal{N}(X \mid \mu_x, \Sigma_{xx}), \quad q_{\mathrm{post}}(X \mid Y) = \mathcal{N}(X \mid \mu_{x|y}, \Sigma_{x|y}), \quad where \tag{4}$$

$$\mu_{x|y} := \mu_x + \Sigma_{xy} \Sigma_{yy}^{-1} (Y - \mu_y), \quad \Sigma_{x|y} := \Sigma_{xx} - \Sigma_{xy} \Sigma_{yy}^{-1} \Sigma_{yx} \tag{5}$$

*where the mean and covariance are matched to the moments of the target density $p(X, Y)$:*

$$\mathbb{E}_p[(X, Y)^T] = \mu, \qquad \mathrm{Cov}_p(X, Y) = \Sigma. \tag{6}$$

Furthermore, the value of $I_{\mathrm{m}+p}$ can be calculated in closed-form as the mutual information of the moment-matched Gaussian (Dahlke et al., 2023). We refer to this as the *Joint Variational Gaussian (JVG)* estimator:

$$I_{\mathrm{JVG}} := H_p(q_{\mathrm{marg}}(X)) - H_p(q_{\mathrm{post}}(X|Y)) = \frac{1}{2} \log |det (2\pi e \Sigma_{xx})| - \frac{1}{2} \log |det (2\pi e \Sigma_{x|y})|. \tag{7}$$

These results ensure that, for Gaussian approximating $q$, the approximate MI $I_{\mathrm{m}+p}$ can be efficiently computed via simple moment-matching operations. The strength of this approach is that it does not require gradient descent and is very fast to compute in practice. The drawback of this approach is that it can only model linear dependence between random variables.

## 2.3 NEURAL VARIATIONAL GAUSSIAN (NVG) ESTIMATOR

To capture nonlinear dependence we relax the assumption that $X$ and $Y$ are jointly Gaussian (Foster et al., 2019). A base Gaussian variational distribution is assumed, but now the mean and variance of the posterior distribution are parameterized by a neural network which is a function of the observation variable $y$:

$$q(X) = \mathcal{N}(X \mid \mu_x, \Sigma_{xx}), \quad q(X \mid Y) = \mathcal{N}(X \mid \mu(Y), \Sigma(Y)) \tag{8}$$

where $\mu(Y)$ and $\Sigma(Y)$ are given by neural network function approximators trained to minimize the bound in Lemma 2.1. Marginal moments $\mu_x$ and $\Sigma_{xx}$ in Eqn. (8) are still learned by moment matching to the true distribution. This results in the *Neural Variational Gaussian (NVG)* estimator:

$$I_{\mathrm{NVG}} := H_p(q(X)) - H_p(q(X|Y)) \approx \frac{1}{2} \log |det (2\pi e \Sigma_{xx})| + \frac{1}{N} \sum_{i=1}^{N} \log \mathcal{N}(x_i | \mu(y_i), \Sigma(y_i)) \tag{9}$$

where $\{(x_i, y_i)\} \sim p$. The relaxed assumption of a Gaussian joint distribution allows for $I_{\mathrm{NVG}}$ to approximate nonlinear dependencies between $X$ and $Y$, however this comes at the computational cost of needing to train a neural network. Furthermore, $I_{\mathrm{NVG}}$ is still constrained in its approximation power by assuming a Gaussian fit for each conditional value as well as the prior.

## 3 FLOW-BASED VARIATIONAL ESTIMATORS

Previous work considered $I_{\mathrm{JVG}}$ and $I_{\mathrm{NVG}}$ as jointly- and conditionally-Gaussian variational approximations for MI, respectively (Dahlke et al., 2023; Foster et al., 2019). Both of these approaches are limited by the expressibility of the Gaussian to an arbitrary distribution. To address this, we introduce a more flexible class of distributions based on normalizing flows. The aim of normalizing flows is to

transform a simple base distribution, typically a Gaussian, via a diffeomorphism to achieve a more expressive distribution. Let $Z \in \mathbb{R}^D$ be a random variable with density $q_Z(Z)$ and let $Z = f(X)$ be a diffeomorphism. Then the density of $X$ is given by the change of basis formula (Bishop & Nasrabadi, 2006):

$$q_X(X) = q_Z(f(X)) \left| det\left(\nabla_x f(X)\right)\right| \tag{10}$$

where $\nabla_x f(X)$ is the Jacobian of $f(X)$ with respect to $X$. The choice of $f(\cdot)$ can be any diffeomorphism, however, to train effectively we must be able to evaluate $f$ and its log determinant efficiently. There have been many proposed flows (Kobyzev et al., 2021) including planar and radial (Rezende & Mohamed, 2016), coupling (Kingma et al., 2016), and auto-regressive flows (Kingma et al., 2016; Papamakarios et al., 2021). For this work we consider rational quadratic auto-regressive flows (Durkan et al., 2019) due to their success in related fields (Kobyzev et al., 2021).

## 3.1 JOINT VARIATIONAL FLOW (JVF) ESTIMATOR

Theorem 2.2 states that moment matching yields optimal parameters of a variational estimator to minimize Lemma 2.1, but only applies for a Gaussian approximation. We introduce a new estimator, based on normalizing flows, that adds flexibility to the variational estimator but that satisfies the moment matching optimality conditions.

**Theorem 3.1** (Moment Matched Flow Distribution). *Let $Z = f(X)$ and $V = g(Y)$ be diffeomorphisms with Gaussian joint density given by:*

$$q_{Z,V}(Z,V) = \mathcal{N}\left(\begin{bmatrix} Z \\ V \end{bmatrix} \middle| \mu := \begin{bmatrix} \mu_z \\ \mu_v \end{bmatrix}, \Sigma := \begin{bmatrix} \Sigma_{zz} & \Sigma_{zv} \\ \Sigma_{vz} & \Sigma_{vv} \end{bmatrix}\right) \tag{11}$$

*then the marginal and posterior optimizing the bound in Lemma 2.1 are given by:*

$$q_{\mathrm{marg}}(X) = \mathcal{N}\left(f(X) \mid \mu_z, \Sigma_{zz}\right) \left| det\left(\nabla_x f(X)\right)\right|,$$
$$q_{\mathrm{post}}(X \mid Y) = \mathcal{N}\left(f(X) \mid \mu_{z|v}, \Sigma_{z|v}\right) \left| det\left(\nabla_x f(X)\right)\right|,$$
$$where \quad \mu_{z|v} := \mu_z + \Sigma_{zv}\Sigma_{vv}^{-1}(g(Y) - \mu_v), \quad \Sigma_{z|v} := \Sigma_{zz} - \Sigma_{zv}\Sigma_{vv}^{-1}\Sigma_{vz} \tag{12}$$

*where the mean and covariance are matched to the moments of the target density $p(X,Y)$:*

$$\mathbb{E}_p[(f(X), g(Y))^T] = \mu, \qquad \mathrm{Cov}_p(f(X), g(Y)) = \Sigma. \tag{13}$$

Theorem 3.1 establishes that the optimal flow distribution is moment-matched to the target distribution, providing a simple solution to parameters of the flow distribution. The joint density on $X$ and $Y$ also takes a convenient form given by the change of basis formula in Eqn. (10):

$$q_{X,Y}(X,Y) = q_{Z,V}(f(X), g(Y)) \left| det\left(\nabla_x f(X)\right)\right| \left| det\left(\nabla_y g(Y)\right)\right| \tag{14}$$

$$= \mathcal{N}\left(\begin{bmatrix} f(X) \\ g(Y) \end{bmatrix} \middle| \mu, \Sigma\right) \left| det\left(\nabla_x f(X)\right)\right| \left| det\left(\nabla_y g(Y)\right)\right| \tag{15}$$

where the Jacobian terms can be easily computed by construction of the flow. What remains is to train parameters of the flows to minimize Lemma 2.1. The following lemma establishes that this bound has a convenient form, which can be optimized via gradient descent.

**Lemma 3.2** (Flow Upper Bound). *Let $p(X,Y)$ be an arbitrary target distribution and $q_{X,Y}(X,Y)$ be the distribution of the form in Eqn. (14) with moment matched flow density $q_{Z,V}(Z,V)$. Then the bound in Lemma 2.1 is given by:*

$$|I_{m+p} - I| \le \frac{1}{2}\log\left|det\left(2\pi e\Sigma_{zz}\right)\right| + \frac{1}{2}\log\left|det\left(2\pi e\Sigma_{z|v}\right)\right| - 2\mathbb{E}_{p_X}\left[\log\left|det\left(\nabla_x f(X)\right)\right|\right] + C \tag{16}$$

We see that the marginal and conditional entropy of $q_{X,Y}$ can be expressed in terms of the underlying Gaussian entropy of $q_{Z,V}(Z,V)$, which has a closed form resulting in the first two terms of Eqn. (16). The last term is the log determinant of the Jacobian, which is easily computed by construction of the normalizing flow. The flows are trained to minimize Eqn. (16) and the parameters $\mu$ and $\Sigma$ are learned from moment matching at each step of the flow training. This results in the *Joint Variational Flow (JVF)* estimator:

$$I_{\mathrm{JVF}} := H_p(q_{\mathrm{marg}}(X)) - H_p(q_{\mathrm{post}}(X|Y)) = \frac{1}{2}\log\left|det\left(2\pi e\Sigma_{zz}\right)\right| - \frac{1}{2}\log\left|det\left(2\pi e\Sigma_{z|v}\right)\right| \tag{17}$$

We see that evaluating the MI does not require the log determinant term introduced by the flow which is the property of MI invariance under invertible transformations (Czyż et al., 2023). While the resulting MI estimator captures only linear dependence (via Gaussian MI) it does so in the flow distribution after application of nonlinear flows. The resulting estimator captures nonlinear dependence in the original $X$ and $Y$.

### 3.1.1 STABLE TRAINING ERROR UPPER BOUND

The bound in Lemma 3.2 only sees the log determinant of $f(X)$, and in practice we found that $g(Y)$ tends to overfit during learning. To address this, we introduce a new bound which is a slight modification to Lemma 2.1.

**Lemma 3.3.** *Let $p(X, Y)$ be any model and $q_{X,Y}(X, Y)$ be of the form in Eqn. (14) with base distribution $q_{Z,V}(Z, V)$, then the following bound holds:*

$$|I_{JVF} - I| \leq 2H_p(q_{Z,V}(Z, V)) - 2\mathbb{E}_{p_X}\left[\log|det\left(\nabla_x f(X)\right)|\right] - 2\mathbb{E}_{p_Y}\left[\log|det\left(\nabla_y g(Y)|\right)\right] + C \tag{18}$$

*For a Gaussian base $q_{Z,V}(Z, V) = \mathcal{N}(\mu, \Sigma)$ the tightest bound of the form Eqn. (18) is given by the moment-matched flow distribution with $\Sigma = \text{Cov}_p(f(X), g(Y))$ and takes the form:*

$$|I_{JVF} - I| \leq \log|det\left(2\pi e\Sigma\right)| - 2\mathbb{E}_{p_X}\left[\log|det\left(\nabla_x f(X)\right)|\right] - 2\mathbb{E}_{p_Y}\left[\log|det\left(\nabla_y g(Y)\right)|\right] + C \tag{19}$$

*where $C = 2H_p(p(X, Y))$. This bound is tight when $q_{X,Y}(X, Y) = p(X, Y)$ almost surely.*

Eqn. (19) contains both the log determinant terms of each flow and in practice enables more stable training. For all experiments in Section 6, we minimize $I_{\text{JVF}}$ with respect to this bound. Calculation of the MI estimator remains unchanged from Eqn. (17). The pseudocode for training JVF can be found in Appendix A.1 in Algorithm 1.

### 3.2 NEURAL VARIATIONAL FLOW (NVF) ESTIMATOR

The estimator $I_{\text{JVF}}$ assumes that $X$ and $Y$ have a joint distribution of the form Eqn. (14). While the underlying joint is non-Gaussian it can still result in restrictions on expressibility. To address this we introduce neural network function approximators to the base (flow) distribution. We add a flow to the marginal, $Z_{\text{marg}} = f_{\text{marg}}(X)$, and to the posterior, $Z_{\text{post}} = f_{\text{post}}(X)$ with the following distributions:

$$q_Z(Z_{\text{marg}}) = \mathcal{N}(\mu_z, \Sigma_{zz}), \qquad q_{Z|Y}(Z_{\text{post}} \mid Y) = \mathcal{N}(\mu(Y), \Sigma(Y)) \tag{20}$$

where $\mu(\cdot)$ and $\Sigma(\cdot)$ are given by neural network function approximators. Then for two normalizing flows, $f_{\text{marg}}(\cdot)$ and $f_{\text{post}}(\cdot)$, the corresponding distributions on $X$ are:

$$q_X(X) = \mathcal{N}(f_{\text{marg}}(X) \mid \mu_z, \Sigma_{zz}) |det\left(\nabla f_{\text{marg}}(X)\right)| \tag{21}$$

$$q_{X|Y}(X \mid Y) = \mathcal{N}(f_{\text{post}}(X) \mid \mu(Y), \Sigma(Y)) |det\left(\nabla f_{\text{post}}(X)\right)| \tag{22}$$

Furthermore, the cross entropy of each distribution is given by:

$$H_p(q_X(X)) = H_{q_Z}(q_Z(Z_{\text{marg}})) - \mathbb{E}_{p_X}\left[\log|det\left(\nabla_x f_{\text{marg}}(X)\right)|\right] \tag{23}$$

$$H_p(q_{X|Y}(X \mid Y)) = \mathbb{E}_{p_X}\left[-\log\mathcal{N}(f_{\text{post}}(X) \mid \mu(Y), \Sigma(Y))\right] - \mathbb{E}_{p_X}\left[\log|det\left(\nabla_x f_{\text{post}}(X)\right)|\right] \tag{24}$$

where $H_{q_Z}(q_Z(Z_{\text{marg}}))$ is the entropy of the underlying Gaussian. The optimal marginal moments $\mu_z$ and $\Sigma_{zz}$ are found via moment matching, but both flows and the posterior parameters are optimized to minimize Lemma 3.3. This results in the *Neural Variational Flow (NVF)* estimator:

$$\begin{aligned} I_{\text{NVF}} = &\frac{1}{2}\log|det\left(2\pi e\Sigma_{zz}\right)| - \frac{1}{N}\sum_{i=1}^{N}\log|det\left(\nabla_x f_{\text{marg}}(x_i)\right)| \\ &+ \frac{1}{N}\sum_{i=1}^{N}\log\mathcal{N}(f_{\text{post}}(x_i) \mid \mu(y_i), \Sigma(y_i)) + \frac{1}{N}\sum_{i=1}^{N}\log|det\left(\nabla_x f_{\text{post}}(x_i)\right)| \end{aligned} \tag{25}$$

where $\{(x_i, y_i)\} \sim p$ and $\Sigma_{zz} = \text{Cov}(f_{\text{marg}}(X))$. The log determinant terms do not cancel in this case, since we apply separate transforms to the marginal and posterior, and so MI is not invariant under these transformations. $I_{\text{NVF}}$ has both the flexibility of a flow-based distribution and nonlinear dependence modeling. $I_{\text{NVF}}$ is capable of estimating much more complex distributions but comes at the cost of needing to train both flows and neural network parameters. The pseudocode for training NVF can be found in Appendix A.1 in Algorithm 2.

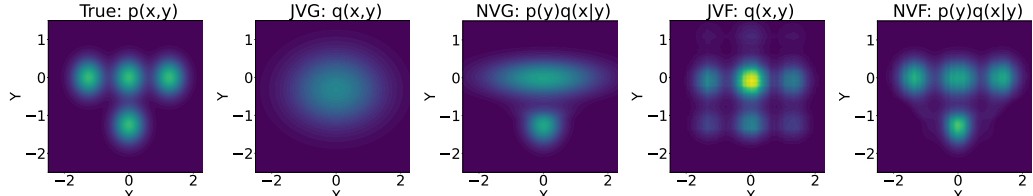

Figure 1: **Multi-modal Density** A four component Gaussian mixture model with nonlinear dependency is used to show the approximating capabilities of each estimator. JVG cannot model the nonlinearity or multi-modality. NVG is able to model the nonlinear dependence between rows, but cannot capture the multi-modality. JVF is able to model the multi-modality across the system but "hallucinates" extra modes to satisfy its linearity constraint. Finally, NVF models the multi-modality and the nonlinear dependency of the GMM.

## 4 PRACTICAL APPLICATION OF VARIATIONAL MI ESTIMATORS

Estimators JVG, NVG, JVF, and NVF have different flexibility incorporated to improve accuracy in estimating MI. Moving from a joint to neural Gaussian increases modeling capability, but comes at the cost of adding a neural network to train and losing access to an explicit joint distribution. Although the joint distribution is not necessary for MI estimation, it can be useful to have a parameterized joint for applications such as generative models and classification. The second flexibility added to the estimators is flows which enhance the estimators capability of capturing multi-modality and nonlinear structures of the distribution but comes at the cost of needing to train two flows.

Figure 1 illustrates these properties on a four-component Gaussian mixture model. We see that JVG can only match the mean and spread of the data from the joint Gaussian assumption. NVG is able to capture the change of dependency from the top three modes to the bottom but is unable to capture the multimodality for any individual $y$. JVF on the other hand is able to model the multi-modality but still is constrained to linear dependence of the transformed variable from the joint assumption. This means that JVF "hallucinates" alternate modes to satisfy its linear dependence constraint. Finally, NVF has both capabilities and therefore is capable of capturing both the multi-modality of the distribution and the non-linearity of the dependence. It is important to note that for both the neural estimators no explicit joint $q(X, Y)$ exists, instead the joint is estimated by $p(Y)q(X \mid Y)$ for illustrative purposes.

The choice of estimator depends on the prior knowledge about the underlying model. JVG is suitable for simple cases, while NVG captures more complex dependencies. JVF is appropriate for multi-modal distributions with linear dependencies, and NVF handles both multi-modality and nonlinear dependencies. Table 1 summarizes the strengths and weaknesses of each estimator to guide the selection of the appropriate method.

Table 1: Flexibility of each estimator. Adding flows increases the variational density flexibility and adding neural parameters increases the dependency flexibility. Adding the neural parameters however removes the modeling of an explicit joint distribution.

| Estimator | Density | Dependency | Training | Joint |
|---|---|---|---|---|
| $I_{\text{JVG}}$ | Inflexible | Inflexible | None | Explicit |
| $I_{\text{NVG}}$ | Inflexible | Flexible | $\mu(Y), \Sigma(Y)$ | Implicit |
| $I_{\text{JVF}}$ | Flexible | Inflexible | $f(X), g(Y)$ | Explicit |
| $I_{\text{NVF}}$ | Flexible | Flexible | $\mu(Y), \Sigma(Y), f_{\text{prior}}(X), f_{\text{post}}(X)$ | Implicit |

### 4.1 BAYESIAN OPTIMAL EXPERIMENTAL DESIGN

A common application of MI is Bayesian Optimal Experimental Design (BOED) (Lindley, 1956) where the goal is to maximize information through a series of decisions, $d$, about a latent variable, $X$, given observations, $Y$. This model consists of an assumed prior on $X$, $p(X)$, and a likelihood function, $p(Y|X, d)$. Each decision is quantified with the amount of information it will likely give, which in the context of BOED is called *expected information gain (EIG)*

$$\text{EIG}(d) = I_d(X, Y) = H_{p(X)}\left[p(X)\right] - H_{p(X, y|d)}\left[p(X|Y, d)\right] \tag{26}$$

The optimal decision is the one that maximizes EIG: $d^* = \text{argmax}_d \text{EIG}(d)$. In sequential experimental design, a sequence of decisions is made $d_1, \ldots, d_T$ and after each decision, an observation is made $Y_1, \ldots, Y_T$. The observations are used to update the prior, $p(X)$, at the next time step using to posterior, $p(X \mid Y_1, \ldots, Y_t, d_1, \ldots, d_t)$, to include the observations from the previous decisions. When making $T$ decisions, the goal is to maximize the *total expected information gain (TEIG)* (Ivanova et al., 2021)

$$\text{TEIG}_T(D) = \mathbb{E}_{p(X)p(h_T|X)} \left[ \sum_{t=1}^{T} \mathbb{E}_{p(X)p(Y_t|X,d_t,h_{t-1})} \left[ \log \frac{p(X \mid Y_t, d_t, h_{t-1})}{p(X \mid h_{t-1})} \right] \right] \quad (27)$$

where $D = \{d_t\}_{t=1}^{T}$ is the set of decisions and $h_t = \{(Y_i, d_i)\}_{i=1}^{T}$ is the *history* of previously taken decisions and their corresponding observations. We utilize $I_{\text{JVG}}$, $I_{\text{NVG}}$, $I_{\text{JVF}}$, or $I_{\text{NVF}}$ as variational estimations of the EIG at each step in a greedy approach to maximizing TEIG. We learn the corresponding parameters for each model simultaneously with the decision which is an approach introduced by Foster et al. (2020) where they utilized the NVG estimator. We highlight this application for the use in Section 6 to show the utility of our estimators in practice.

## 5 RELATED WORK

The base MI approximation, $I_{\text{m}+p}$, considered in this work is one of many possible variational approximations. Choosing only one distribution to approximate with a variational distribution results in a bound of MI. These are considered in alternative work (Foster et al., 2019; Dahlke et al., 2023; Poole et al., 2019) but predominantly utilize only Gaussians as the variational distribution. Canonical correlation analysis (CCA) assumes a joint Gaussian variational distribution which is closely related to $I_{\text{JVG}}$. (Cheng et al., 2020) proposed an upper bound of MI that utilizes a variational distribution similar to that of $I_{\text{NVG}}$. Recent work from (Butakov et al., 2024) has proposed a similar approach also utilizing normalizing flows which is closely related to $I_{\text{JVF}}$, but does not consider an estimator analogous to $I_{\text{NVF}}$. Furthermore, (Dong et al., 2025) considers using conditional normalizing flows in the context of sequential BOED.

An alternative approach to approximating the distributions is to instead use a discriminator. The Donsker-Varadhan representation and the corresponding f-divergence representation lead to tight lower bounds on MI, resulting in a variety of approaches: DV (Donsker & Varadhan, 1975), MINE (Belghazi et al., 2018), and NWJ (Nguyen et al., 2010). Other methods use noise-contrastive estimation of the density ratio in MI which leads to a lower-bound of MI, called InfoNCE (van den Oord et al., 2019). Approaches that do not assume a discriminator or a density but instead compare $k$-nearest neighbors of samples have been considered called KSG (Kraskov et al., 2004). For a thorough review of MI approximations see (Poole et al., 2019).

In the space of BOED multiple approaches have been taken to select a series of actions that maximizes MI. Multiple greedy approaches have been considered where each design is chosen to maximize instantaneous EIG. The approach of simultaneously optimizing the variational distributions and decision via gradient decent has been considered (Foster et al., 2020). Other approaches, such as LFIRE (Kleinegesse et al., 2021), perform ratio estimation of MI with Bayesian optimization for the decisions. Batch optimization is the approach where all decisions are made before testing time and stay fixed regardless of the observed data, such as MINEBED (Kleinegesse & Gutmann, 2020). Finally, recent work takes a reinforcement learning approach where a policy-discriminator pair is learned (Foster et al., 2021; Ivanova et al., 2021; Blau et al., 2022; Huan & Marzouk, 2016). The discriminator plays the role of MI estimation for a select sequence of decision and observations while the policy is trained to make informative decisions based upon its history of decisions and observations.

## 6 EXPERIMENTS

We consider a wide range of experiments, from synthetic to application based. We start by comparing our MI estimation capabilities against many common estimators: DV (Donsker & Varadhan, 1975), MINE (Belghazi et al., 2018), NWJ (Nguyen et al., 2010), InfoNCE (van den Oord et al., 2019), CCA (Murphy, 2023), and KSG (Kraskov et al., 2004). The benchmarking tests are a large MI estimation

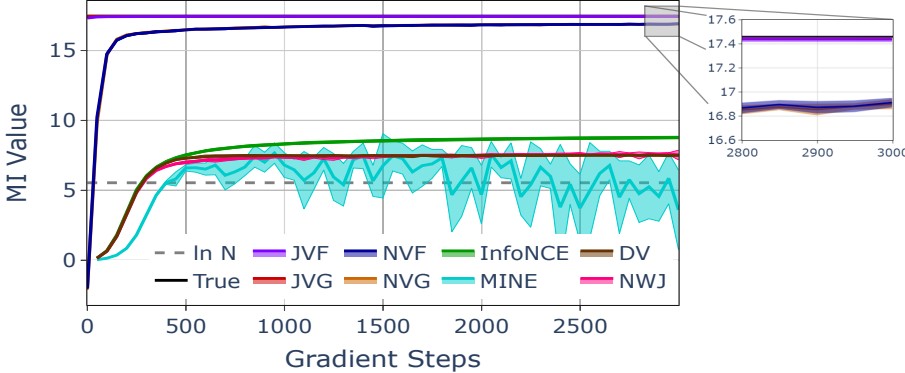

Figure 2: **Large MI** is difficult for many estimators to approximate. We see that the four distribution based estimators, $I_{\text{JVG}}$, $I_{\text{NVG}}$, $I_{\text{JVF}}$, and $I_{\text{NVF}}$, can accurately predict MI, but the four discriminative-based methods, MINE, InfoNCE, NWJ, and DV, all converge around $\mathcal{O}\log(N)$.

(McAllester & Stratos, 2020) and a diverse set of 36 individual distributions with known ground truth MI values (Czyż et al., 2023). We then look at the sequential decision making setting of location finding. All experiments were run on a high-performance computing cluster with nodes consisting of 2x AMD EPYC 7642 48-core (Rome) CPUs, 512GB of RAM, and NVIDIA V100S GPUs.

## 6.1 HIGH MUTUAL INFORMATION EXPERIMENT

We begin with a well-known difficult case of estimating large MI. McAllester & Stratos (2020) showed that any distribution-free high-confidence lower bound on MI, such as the commonly used discriminative-based methods, estimated from $N$ samples cannot be larger than $\mathcal{O}(\log N)$. We create a Gaussian where $X, Y \in \mathbb{R}^{15}$ with a correlation $\rho = .95$. The MI of this distribution can be exactly computed as: $I(X, Y) = -\frac{Dim_X}{2} \ln(1 - \rho^2) = 17.459$. We take a large set of $75,000$ samples to train our distribution based estimators as well as a variety of discriminative based estimators for 3000 training steps using a batch size of $N = 256$. We utilize a 80-20 split of training and testing samples of the total samples. Figure 2 shows the best testing value for each estimator. We see that $I_{\text{JVG}}$ and $I_{\text{JVF}}$ converge nearly instantaneously to the true MI value and $I_{\text{NVG}}$ and $I_{\text{NVF}}$ both converge rapidly to high-quality estimates. For the discriminative-based methods, we see that they learn slower and converge to values drastically lower than the true MI.

## 6.2 MUTUAL INFORMATION BENCHMARK

Our next experiment is a collection of benchmarks from Czyż et al. (2023). They construct a diverse family of distributions with known ground-truth MI consisting of Gaussian, Uniform, and Student-T distributions that have MI invariant transformations applied to them (see Appendix B.2 for more details). We use $N = 1,000$ samples per dimension, with a train-test split of 50-50. Figure 3 displays the average MI over 10 runs for each experiment. Due to the large number of experiments we focus on a selection here (see Appendix B.2 for all 36). For the majority of experiments we see that incorporating flow improves estimation accuracy, with JVF generally outperforming JVG. In the neural estimator case (NVF vs. NVG) the relative improvement is more limited, with a few important exceptions discussed next.

**Challenging cases** for all methods are the **spiral multinormal** and the **student-t distributions**. The spiral multinormal MI is substantially underestimated by all estimators, but we see that flow-based methods (JVF, NVF) typically outperform their non-flow counterparts (JVG, NVG) as well as the distribution-free baselines. For the student-t distributions, all methods produce unstable estimates due to high-variance (due to space we omit standard errors). The high variance for both JVG and NVG comes from the fact that for many cases moments of the student-t distribution are infinite or undefined, limiting effectiveness of moment-matching steps required by JVG and NVG. JVF and NVF can potentially address this issue with the flows, allowing the points to have finite moments, however this is particularly difficult to adequately transform the fat tails of the student-t with sparse samples. Czyż et al. (2023) introduce the **inverse hyperbolic sin transformation** as a means of normalizing the

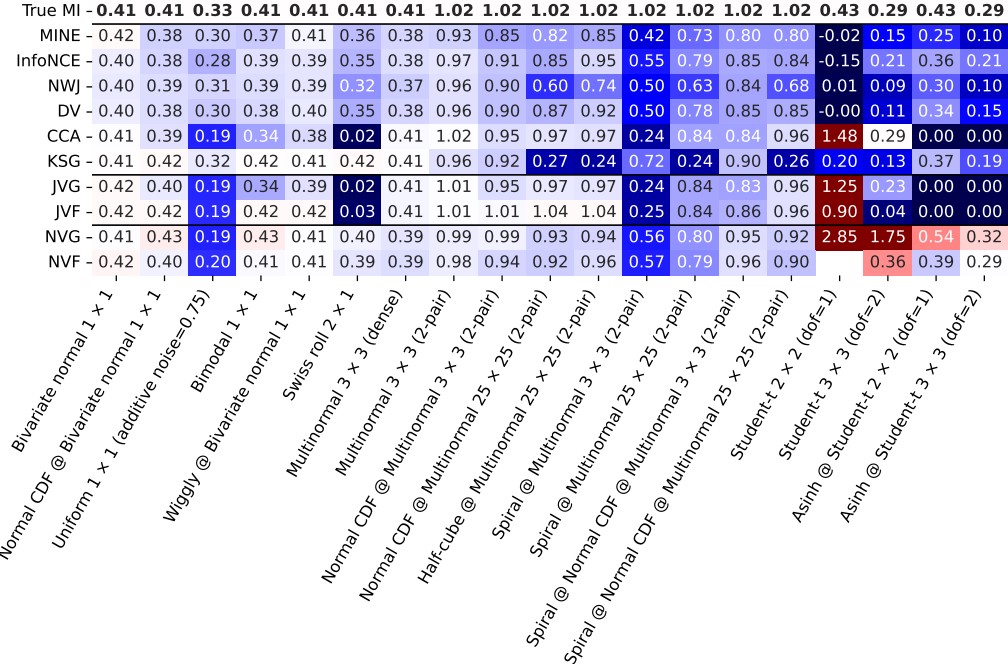

Figure 3: **MI Benchmark** runs against a selection of 19 different scenarios (for space; c.f. Appendix for all 36). Blue indicates an underestimate and red indicates on overestimate with the magnitude indicated by saturation. Blank cells indicate that the method produced divergent estimates.

student-t tails, and thus creating finite moments. In this case we see $I_{\text{NVF}}$ produces good results of the MI, where all other methods seem to fail, suggesting the estimator is useful for fat tailed distributions with appropriate data regularization. Finally, we notice the **uniform additive noise** as the only case which the discriminative-based methods outperform distribution-based methods.

### 6.3 LOCATION FINDING

We now demonstrate our methods in application to BOED discussed in Section 4.1 on the location finding experiment (Ivanova et al., 2021). In this experiment there are multiple sources with unknown locations, $X$, producing a signal whose intensity decreases according to the inverse square law. The signals from all sources are summed together to create a total intensity, $Y$ that can be nosily observed from any location, $d$. Our specific setting contains two sources in two dimensions, for a total of a four dimensional $X \in \mathbb{R}^4$ and a one dimensional observation, $Y \in \mathbb{R}^1$. The goal is to make a sequence of $T = 10$ decisions $d_1, \ldots, d_T \in \mathbb{R}^2$ that maximize the total expected information gain (Eqn. (27)). We consider iDAD (Ivanova et al., 2021) as a strong baseline in this experiment as iDAD performs non-myopic decision making and so should represent an upper bound on what is achievable by the greedy (myopic) methods JVG, JVG, NVG, and NVF. iDAD also requires more total computation time as it performs a significant offline training phase that other methods do not.

For our main results we see that incorporating flow uniformly improves MI estimation. When using the correct prior $p(X) = \mathcal{N}(0, I)$ the JVG estimator has the lowest MI estimate due to assumption of a joint Gaussian distribution on $X$ and $Y$. Incorporating flow (JVF) shows a significant improvement. The neural estimators show more flexibility but we see similar improvements when incorporating flow (NVF) to the NVG estimator. All greedy methods require less total computation time than the iDAD. Note that neural methods benefit from GPU acceleration that was not applied in our implementation of JVG or JVF. One drawback of ammortized methods, such as iDAD, is that they do not adapt to model mismatch during the test phase; a result of the well-known *ammortization gap* (Cremer et al., 2018). We test robustness of each method by using a mismatched prior in the test phase $p(X) = \mathcal{N}(1, I)$. The relative performance of all methods remains identical to the matched prior setting. But iDAD exhibits a 30% reduction in MI estimate whereas the greedy methods, which

Table 2: **Total Expected Information Gain** is reported for two settings of the location finding experiment, *matched prior* where the training prior matches the true prior and *mismatched prior* where the testing and training priors differ. We see that JVG, JVF, and NVG all fail to make informative decisions due to their limited flexibility and greedy decision making in both settings. We notice that iDAD performs well in the setting where its training data matches the testing data, but suffers a substantial amortization cost as the testing values are shifted. NVF yields informative decisions while being comparatively robust to model mismatch. Total computation time is lower for the variational methods when accounting for the offline training required by iDAD.

| Method | Matched Prior | Mismatched Prior | Training Time | Deployment Time (s) |
|---|---|---|---|---|
| Random | $4.62 \pm 0.24$ | $3.59 \pm 0.14$ | - | - |
| iDAD (InfoNCE) | $7.54 \pm 0.19$ | $5.29 \pm 0.21$ | 8826.683 | $0.0256 \pm 0.0016$ |
| JVG | $3.56 \pm 0.19$ | $3.59 \pm 0.15$ | - | $287 \pm 35$ |
| JVF | $4.30 \pm 0.19$ | $3.85 \pm 0.19$ | - | $1730 \pm 207$ |
| NVG | $4.61 \pm 0.19$ | $3.99 \pm 0.20$ | - | $186 \pm 21$ |
| NVF | $5.10 \pm 0.20$ | $4.748 \pm 0.18$ | - | $1253 \pm 155$ |

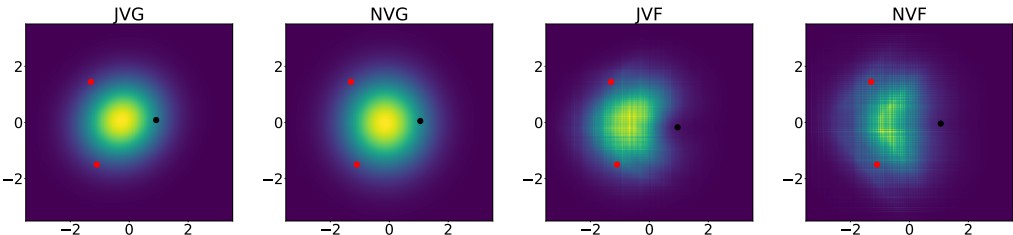

Figure 4: **Location Finding Posterior** estimations after a single observation. We notice that JVG, NVG both estimate the posterior to be a Gaussian. The introduction of flow to JVF and NVF increases flexibility to meaningfully approximate the posterior.

perform inference at test time either show no reduction in MI (JVG) or a smaller relative decrease (7%-13%).

We notice that in this experiment JVG, NVG, and JVF perform significantly worse than NVF. To shed some light on this, we can look at the posterior prediction after making a single observation (Fig.4). The result of this observation increases the belief that sources are located in a radius away from the origin. Neither JVG nor NVG are flexible enough to capture this belief and instead produce Gaussian posteriors. Including flow JVF and NVF have increasesed flexibility to capture the non-Gaussian belief. We see that the additional degrees of freedom in NVF allows for a higher posterior belief in the vicinity of the true target locations, whereas JVF concentrates more away from the targets.

## 7 DISCUSSION

We introduced two flow-based variational estimators, $I_{\text{JVF}}$ and $I_{\text{NVF}}$, which are built on previous Gaussian estimators, $I_{\text{JVG}}$ and $I_{\text{NVG}}$. The inclusion of flows increases the expressibility of the variational estimators. These estimators in general are just as accurate as widely used critic-based estimators, but are not limited at their capability of learning large MI values. Furthermore, $I_{\text{NVF}}$ has the capability of estimating wide-tale distribution MI given appropriate data regularization, which are notoriously difficult distributions to estimate. Finally, we find that these methods are effective for sequential Bayesian experimental design. In the BOED setting these methods show robustness to model mismatch not exhibited by policy-based approaches while also yielding significantly lower computation time when training is accounted for.

**Limitations**   Our proposed flow-based estimators have increased the flexibility of the distribution based variational MI estimators. However, there are still a few cases where we are unable to accurately model MI, such as additive noise in Section 6.2. Furthermore, each method adds increased number of parameters to train which can be costly, specifically at high dimensions. Future work plans to consider ways to increase training speed or remove the necessity of including neural networks.

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

## A  APPENDIX

### A.1  PSEUDOCODE

We briefly provide an outline of each methods pseudo code. So of the notation is lightly changed in an attempt to highlight parameters of models. For example, for NVF, $f_{\text{prior}}(X)$ is referred to as $f_{\theta_{\text{prior}}}(X)$ to highlight the parameters, $\theta_{\text{prior}}$, of the prior flow.

**for** $i=1:K$ **do**

> Sample $\{x_j\}_{j=1:N} \sim p(X)$;
> Sample $\{y_j\}_{j=1:N} \sim p(y \mid x_j)$;
> $\mu \leftarrow \frac{1}{N} \sum_{j=1}^{N} (f_\theta(x_j)^T, g_\psi(y_j)^T)^T$;
> $\Sigma \leftarrow \frac{1}{N-1} \sum_{j=1}^{N} (f_\theta(x_j)^T, g_\psi(y_j)^T)(f_\theta(x_j)^T, g_\psi(y_j)^T)^T - \mu\mu^T$;
> Loss $\leftarrow \log |2\pi e\Sigma| - \frac{2}{N} \sum_{j=1}^{N} \log |\nabla_x f_\theta(x_j)| - \frac{2}{N} \sum_{j=1}^{N} \log |\nabla_y g_\psi(y_j)|$;
> $\theta \leftarrow \theta - \alpha \nabla_\theta \text{Loss}$;
> $\psi \leftarrow \psi - \beta \nabla_\psi \text{Loss}$;

**end**

$\mu \leftarrow \frac{1}{N} \sum_{j=1}^{N} (f_\theta(x_j)^T, g_\psi(y_j)^T)^T$;
$\Sigma \leftarrow \frac{1}{N-1} \sum_{j=1}^{N} (f_\theta(x_j)^T, g_\psi(y_j)^T)(f_\theta(x_j)^T, g_\psi(y_j)^T)^T - \mu\mu^T$;
$I_{JVF} \leftarrow \frac{1}{2} \log |2\pi e\Sigma_Z| - \frac{1}{2} \log \left|2\pi e\Sigma_{Z|V}\right|$

**Algorithm 1:** JVF Pseudocode

**for** $i=1:K$ **do**

> Sample $\{x_j\}_{j=1:N} \sim p(X)$;
> Sample $\{y_j\}_{j=1:N} \sim p(y \mid x_j)$;
> $\mu_Z \leftarrow \frac{1}{N} \sum_{j=1}^{N} f_{\theta_{\text{prior}}}(x_j)$;
> $\Sigma_{ZZ} \leftarrow \frac{1}{N-1} \sum_{j=1}^{N} f_{\theta_{\text{prior}}}(x_j) f_{\theta_{\text{prior}}}(x_j)^T - \mu_Z \mu_Z^T$;
> Loss $\leftarrow \log |2\pi e\Sigma_Z Z| - \frac{1}{N} \sum_{j=1}^{N} \log \left|\nabla_x f_{\theta_{\text{prior}}}(x_j)\right| -$
> $\quad \frac{1}{N} \sum_{j=1}^{N} \left(\log \mathcal{N}\left(f_{\theta_{\text{post}}}(x_j) \mid \mu_\phi(y_i), \Sigma_\phi(y_i)\right) + \log \left|\nabla_x f_{\theta_{\text{post}}}(x_j)\right|\right)$;
> $\theta_{\text{prior}} \leftarrow \theta_{\text{prior}} - \alpha \nabla_{\theta_{\text{prior}}} \text{Loss}$;
> $\theta_{\text{post}} \leftarrow \theta_{\text{post}} - \alpha \nabla_{\theta_{\text{post}}} \text{Loss}$;
> $\phi \leftarrow \phi - \beta \nabla_\phi \text{Loss}$;

**end**

$\mu_Z \leftarrow \frac{1}{N} \sum_{j=1}^{N} f_{\theta_{\text{prior}}}(x_j)$;
$\Sigma_{ZZ} \leftarrow \frac{1}{N-1} \sum_{j=1}^{N} f_{\theta_{\text{prior}}}(x_j) f_{\theta_{\text{prior}}}(x_j)^T - \mu_Z \mu_Z^T$;
$I_{JVF} \leftarrow \log |2\pi e\Sigma_Z Z| - \frac{1}{N} \sum_{j=1}^{N} \log \left|\nabla_x f_{\theta_{\text{prior}}}(x_j)\right| +$
$\quad \frac{1}{N} \sum_{j=1}^{N} \left(\log \mathcal{N}\left(f_{\theta_{\text{post}}}(x_j) \mid \mu_\phi(y_i), \Sigma_\phi(y_i)\right) + \log \left|\nabla_x f_{\theta_{\text{post}}}(x_j)\right|\right)$

**Algorithm 2:** NVF Pseudocode

A.2 SECTION 2 PROOFS

**Lemma 2.1.** *For any model $p(X, Y)$ and distributions $q_{\mathrm{marg}}(X)$, $q_{\mathrm{post}}(X \mid Y)$, the following holds:*

$$|I_{m+p} - I| \leq \min_{q_{\mathrm{marg}}} H_p(q_{\mathrm{marg}}(X)) + \min_{q_{\mathrm{post}}} H_p(q_{\mathrm{post}}(X \mid Y)) + C$$

*where $C = -H_p(p(X)) - H_p(p(X \mid Y))$ does not depend on $q_{\mathrm{marg}}$ or $q_{\mathrm{post}}$. Further, the RHS is $0$ iff $q_{\mathrm{marg}}(X) = p(X)$ and $q_{\mathrm{post}}(X \mid Y) = p(X \mid Y)$ almost surely.*

*Proof.* We recreate the proof from Foster et al. (2019)

$$|I_{\mathrm{m}+p} - I| = |H_p(q_{\mathrm{marg}}(X)) - H_p(q_{\mathrm{post}}(X \mid Y)) - H_p(p(X)) + H_p(p(X \mid Y))| \tag{28}$$

$$= |-H_p(p(X)) + H_p(q_{\mathrm{marg}}(X)) + H_p(p(X \mid Y)) - H_p(q_{\mathrm{post}}(X \mid Y))| \tag{29}$$

$$= |\mathrm{KL}(p(X) \,\|\, q_{\mathrm{marg}}(X)) - \mathrm{KL}(p(X \mid Y) \,\|\, q_{\mathrm{post}}(X \mid Y))| \tag{30}$$

$$\leq |\mathrm{KL}(p(X) \,\|\, q_{\mathrm{marg}}(X))| + |\mathrm{KL}(p(X \mid Y) \,\|\, q_{\mathrm{post}}(X \mid Y))| \tag{31}$$

$$= -H_p(p(X)) + H_p(q_{\mathrm{marg}}(X)) - H_p(p(X \mid Y)) + H_p(q_{\mathrm{post}}(X \mid Y)) \tag{32}$$

$$= H_p(q_{\mathrm{marg}}(X)) + H_p(q_{\mathrm{post}}(X \mid Y)) + C \tag{33}$$

where $C = -H_p(p(X)) - H_p(p(X \mid Y))$ $\qquad\square$

To prove Theorem 2.2 and Theorem 3.1, we rely on results from Dahlke et al. (2023). In Section A.4, we include the related theorems and proofs necessary for the results of this paper as a convenience to the reader.

**Theorem 2.2** (Moment Matching = Optimization). *Let $q(X, Y)$ be a joint Gaussian density:*

$$q(X, Y) = \mathcal{N}\left( \begin{bmatrix} X \\ Y \end{bmatrix} \middle| \mu := \begin{bmatrix} \mu_x \\ \mu_y \end{bmatrix}, \Sigma := \begin{bmatrix} \Sigma_{xx} & \Sigma_{xy} \\ \Sigma_{yx} & \Sigma_{yy} \end{bmatrix} \right) \tag{34}$$

*then the marginal and posterior optimizing the bound in Lemma 2.1 are given by:*

$$q_{\mathrm{marg}}(X) = \mathcal{N}\left(X \mid \mu_x, \Sigma_{xx}\right), \quad q_{\mathrm{post}}(X \mid Y) = \mathcal{N}(X \mid \mu_{x|y}, \Sigma_{x|y}), \quad where \tag{35}$$

$$\mu_{x|y} := \mu_x + \Sigma_{xy}\Sigma_{yy}^{-1}(y - \mu_Y), \quad \Sigma_{x|y} := \Sigma_{xx} - \Sigma_{xy}\Sigma_{yy}^{-1}\Sigma_{yx} \tag{36}$$

*where the mean and covariance are matched to the moments of the target density $p(X, Y)$:*

$$\mathbb{E}_p[(X, Y)^T] = \mu, \qquad \mathrm{Cov}_p(X, Y) = \Sigma. \tag{37}$$

*Proof.* It suffices to verify the assumption of the posterior expected statistics being a linear combination of joint statistics (Eqn. (97)) is satisfied. Recall the sufficient statistics of a multivariate Gaussian

$$T(X, Y) = \begin{bmatrix} X \\ Y \\ \mathrm{vec}(XX^T) \\ \mathrm{vec}(XY^T) \\ \mathrm{vec}(YY^T) \end{bmatrix}$$

In this case $\tau_1(Y) = y$ and $\tau_2(Y) = \mathrm{vec}(yy^T)$. We now verify that the expected value under $q_{\mathrm{post}}(X \mid Y)$ of each term in the sufficient statistic is a linear function of $\tau_1(Y)$ and $\tau_2(Y)$

1. $x$

$$\mathbb{E}_{q_{\mathrm{post}}(X|Y))}[x] = \mu_{x|y} = \mu_x + \Sigma_{xy}\Sigma_{yy}^{-1}(y - \mu_Y)$$

2. $y$

$$\mathbb{E}_{q_{\mathrm{post}}(X|Y)}[y] = y$$

3. $xx^T$

$$\mathbb{E}_{q_{\mathrm{post}}(X|Y)}\left[xx^T\right] = \Sigma_{x|y} + \mu_{x|y}\mu_{x|y}^T$$

$$= \Sigma_{xx} - \Sigma_{xy}\Sigma_{yy}\Sigma_{xy}^T + (\mu_x + \Sigma_{xy}\Sigma_{yy}^{-1}(y - \mu_Y))(\mu_x + \Sigma_{xy}\Sigma_{yy}^{-1}(y - \mu_Y))^T$$

$$= \Sigma_{xx} - \Sigma_{xy}\Sigma_{yy}\Sigma_{xy}^T + \mu_x\mu_x^T + \dots$$

$$\mu_x(y - \mu_y^T)\Sigma_{yy}^{-1}\Sigma_{xy}^T + \Sigma_{xy}\Sigma_{yy}^{-1}(y - \mu_Y)\mu_x + \dots$$

$$\Sigma_{xy}\Sigma_{yy}^{-1}(y - \mu_Y)(y - \mu_Y)^T\Sigma_{yy}^{-1}\Sigma_{xy}^T$$

4. $xy^T$

$$
\begin{aligned}
\mathbb{E}_{q_{\text{post}}(X|Y)}\left[xy^T\right] &= (\mu_x + \Sigma_{xy}\Sigma_{yy}^{-1}(y - \mu_Y))y^T \\
&= (\mu_x - \Sigma_{xy}\Sigma_{yy}^{-1}\mu_Y)y^T + \Sigma_{xy}\Sigma_{yy}^{-1}yy^T
\end{aligned}
$$

5. $yy^T$

$$
\mathbb{E}_{q_{\text{post}}(X|Y)}\left[yy^T\right] = yy^T
$$

So the statistics are linear functions of $\tau_1(Y) = y$ and $\tau_2(Y) = yy^T$ so $q_{\text{post}}(X \mid Y)$ satisfies the conditions of Theorem A.5 and moment matching the joint $q(X, Y) = \mathcal{N}(m, \Sigma)$ yields the optimal $q_{\text{post}}(X \mid Y)$. Furthermore, the joint Gaussian sufficient statistics are

$$
T(X, Y) = \left[X, Y, \text{vec}(XX^T), \text{vec}(XY^T), \text{vec}(YY^T)\right]^T
$$

and the sufficient statistics of a marginal distribution are

$$
T(X) = \left[X, \text{vec}(XX^T)\right]^T
$$

which are simply the first and third sufficient statistic from the joint. Therefore moment matching the joint Gaussian trivially moment matches the marginal, giving the optimal $q_{\text{marg}}(X)$. $\qquad\square$

Moment matching the joint Gaussian yields the optimal $q_{\text{marg}}$ and $q_{\text{post}}$. With this we can derive the formula for the MI estimate $I_{\text{JVG}}$. We utilize another results from Dahlke et al. (2023)

**Lemma A.1.** *Analytic Entropy*
*Let $p(X)$ be any distribution and $q(X)$ be in the exponential family with constant base measure, $h(X) = C$, which is analytically moment matched to $p(X)$ and $\hat{q}(X)$ is empirically moment matched, then*

$$
H_p(q(X)) = H_q(q(X)) \quad \hat{H}_p(\hat{q}(X)) = H_{\hat{q}}(\hat{q}(X)) \tag{38}
$$

Lemma A.1 allow us to replace the cross entropy terms with simply the entropy of the corresponding Gaussian, allowing for us to have a closed form equation for the JVG estimation

$$
I_{\text{JVG}}(X, Y) := H_p(q_{\text{marg}}(X)) - H_p(q_{\text{post}}(X \mid Y)) \tag{39}
$$

$$
= H_{q_{\text{marg}}}(q_{\text{marg}}(X)) - H_{q_{\text{post}}}(q_{\text{post}}(X \mid Y)) = \frac{1}{2}\log|2\pi e\Sigma_x| - \frac{1}{2}\log|2\pi e\Sigma_{x|y}| \tag{40}
$$

### A.3 SECTION 3 PROOFS

**Theorem 3.1** (Moment Matched Flow Distribution). *Let $Z = f(X)$ and $V = g(Y)$ be diffeomorphisms with Gaussian joint density given by:*

$$
q_{Z,V}(Z, V) = \mathcal{N}\left(\begin{bmatrix} z \\ v \end{bmatrix} \middle| \mu := \begin{bmatrix} \mu_z \\ \mu_v \end{bmatrix}, \Sigma := \begin{bmatrix} \Sigma_{zz} & \Sigma_{zv} \\ \Sigma_{vz} & \Sigma_{vv} \end{bmatrix}\right) \tag{41}
$$

*then the marginal and posterior optimizing the bound in Lemma 2.1 are given by:*

$$
q_{\text{marg}}(X) = \mathcal{N}\left(f(X) \mid \mu_z, \Sigma_{zz}\right)|\nabla_x f(X)|, \quad q_{\text{post}}(X \mid Y) = \mathcal{N}\left(f(X) \mid \mu_{z|v}, \Sigma_{z|v}\right)|\nabla_x f(X)|,
$$

$$
\text{where} \quad \mu_{z|v} := \mu_z + \Sigma_{zv}\Sigma_{vv}^{-1}(g(Y) - \mu_v), \quad \Sigma_{z|v} := \Sigma_{zz} - \Sigma_{zv}\Sigma_{vv}^{-1}\Sigma_{vz} \tag{42}
$$

*where the mean and covariance are matched to the moments of the target density $p(X, Y)$:*

$$
\mathbb{E}_p[(f(X), g(Y))^T] = \mu, \qquad \text{Cov}_p(f(X), g(Y)) = \Sigma. \tag{43}
$$

*Proof.* Let $Z = f(X)$ and $V = g(Y)$ be diffeomorphisms with Gaussian joint density, then the joint in $X$ and $Y$ can be found by a simple change of variable (Bishop & Nasrabadi, 2006).

$$
q_{X,Y}(X, Y) = \mathcal{N}\left(\begin{bmatrix} f(X) \\ g(Y) \end{bmatrix} \middle| \mu, \Sigma\right)|\nabla_x f(X)|\,|\nabla_y g(Y)| \tag{44}
$$

We now must show that $q_{X,Y}(X,Y)$ satisfies the moment matching condition

$$\mathbb{E}_{q_{X|Y}(X|Y)}[T(X,Y)] = \sum_i g_i(\eta)\tau_i(Y)$$

$q_{X,Y}(X,Y)$ is in the exponential family with base measure $h(X,Y) = |\nabla_x f(X)| |\nabla_y g(Y)|$ and $T(X,Y) = \left[f(X), g(Y), \text{vec}(f(X)f(X)^T), \text{vec}(f(X)g(Y)^T), \text{vec}(g(Y)g(Y)^T)\right]$ which are the sufficient statistics we must consider for Theorem A.5. Furthermore, we can derive the posterior

$$q_{X|Y}(X \mid Y) = \frac{q_{X,Y}(X,Y)}{q_Y(Y)} = \frac{\mathcal{N}\left(\begin{bmatrix} f(X) \\ g(Y) \end{bmatrix} \middle| \mu, \Sigma\right) |\nabla_x f(X)| |\nabla_y g(Y)|}{\int \mathcal{N}\left(\begin{bmatrix} f(X) \\ g(Y) \end{bmatrix} \middle| \mu, \Sigma\right) |\nabla_x f(X)| |\nabla_y g(Y)| \, dx}$$

$$= \frac{\mathcal{N}\left(\begin{bmatrix} f(X) \\ g(Y) \end{bmatrix} \middle| \mu, \Sigma\right) |\nabla_x f(X)|}{\int \mathcal{N}\left(\begin{bmatrix} f(X) \\ g(Y) \end{bmatrix} \middle| \mu, \Sigma\right) |\nabla_x f(X)| \, dx}$$

$$= \frac{\mathcal{N}\left(\begin{bmatrix} f(X) \\ g(Y) \end{bmatrix} \middle| \mu, \Sigma\right) |\nabla_x f(X)|}{\int \mathcal{N}\left(\begin{bmatrix} z \\ g(Y) \end{bmatrix} \middle| \mu, \Sigma\right) dz}$$

$$= \frac{\mathcal{N}\left(\begin{bmatrix} f(X) \\ g(Y) \end{bmatrix} \middle| \mu, \Sigma\right) |\nabla_x f(X)|}{\mathcal{N}\left(g(Y) \middle| \mu_V, \Sigma_{VV}\right)}$$

$$= \mathcal{N}\left(f(X) \middle| \mu_{Z|V}, \Sigma_{Z|V}\right) |\nabla_x f(X)|$$

where $\mu_{Z|V} = \mu_Z + \Sigma_{ZV}\Sigma_{VV}^{-1}(v - \mu_V) = \mu_Z + \Sigma_{ZV}\Sigma_{VV}^{-1}(g(Y) - \mu_V)$ and $\Sigma_{Z|V} = \Sigma_{ZZ} - \Sigma_{ZV}\Sigma_{VV}^{-1}\Sigma_{VZ}$. Using these, we can easily check the linearity condition of each $T(X,Y)$.

1. $f(X)$

$$\mathbb{E}_{q_{X|Y}(X|Y)}[f(X)] = \int \mathcal{N}\left(f(X) \middle| \mu_{Z|V}, \Sigma_{Z|V}\right) |\nabla_x f(X)| f(X) dx$$

$$= \int \mathcal{N}\left(Z \middle| \mu_{Z|V}, \Sigma_{Z|V}\right) z \, dz$$

$$= \mu_{Z|V} = \mu_Z + \Sigma_{ZV}\Sigma_{VV}^{-1}(g(Y) - \mu_V)$$

2. $g(Y)$

$$\mathbb{E}_{q_{X|Y}(X|Y)}[g(Y)] = g(Y)$$

3. $f(X)f(X)^T$

$$\mathbb{E}_{q_{X|Y}(X|Y)}\left[f(X)f(X)^T\right] = \int \mathcal{N}\left(f(X) \middle| \mu_{Z|V}, \Sigma_{Z|V}\right) |\nabla_x f(X)| f(X)f(X)^T dx$$

$$= \int \mathcal{N}\left(Z \middle| \mu_{Z|V}, \Sigma_{Z|V}\right) zz^T dz$$

$$= \Sigma_{Z|V} + \mu_{Z|V}\mu_{Z|V}^T$$

$$= \Sigma_{ZZ} - \Sigma_{Zv}\Sigma_{VV}^{-1}\Sigma_{ZV}^T + \dots$$

$$(\mu_Z + \Sigma_{ZV}\Sigma_{VV}^{-1}(g(Y) - \mu_V))(\mu_Z + \Sigma_{ZV}\Sigma_{VV}^{-1}(g(Y) - \mu_V))^T$$

$$= \Sigma_{ZZ} - \Sigma_{ZV}\Sigma_{VV}^{-1}\Sigma_{ZV}^T + \mu_Z\mu_Z^T + \dots$$

$$\mu_Z(g(Y) - \mu_V^T)\Sigma_{VV}^{-1}\Sigma_{ZV}^T + \Sigma_{ZV}\Sigma_{VV}^{-1}(g(Y) - \mu_V)\mu_Z + \dots$$

$$\Sigma_{ZV}\Sigma_{VV}^{-1}(g(Y) - \mu_V)(g(Y) - \mu_V)^T\Sigma_{VV}^{-1}\Sigma_{ZV}^T$$

4. $f(X)g(Y)^T$

$$
\begin{aligned}
\mathbb{E}_{q_{X|Y}(X|Y)}\left[f(X)g(Y)^T\right] &= \mathbb{E}_{q_{X|Y}(X|Y)}\left[f(X)\right]g(Y)^T \\
&= \left(\mu_Z + \Sigma_{ZV}\Sigma_{VV}^{-1}(g(Y)-\mu_V)\right)g(Y)^T \\
&= \left(\mu_Z - \Sigma_{ZV}\Sigma_{VV}^{-1}\mu_V\right)g(Y)^T + \Sigma_{ZV}\Sigma_{VV}^{-1}g(Y)g(Y)^T
\end{aligned}
$$

5. $g(Y)g(Y)^T$

$$
\mathbb{E}_{q_{X|Y}(X|Y)}\left[g(Y)g(Y)^T\right] = g(Y)g(Y)^T
$$

Since these are all linear functions of $\tau_1(Y) = g(Y)$ and $\tau_2(Y) = g(Y)g(Y)^T$, then $q_{X|Y}(X \mid Y)$ satisfies the conditions of Theorem A.5 and moment matching the joint $q_{Z,V}(Z,V) = \mathcal{N}(m,\Sigma)$ yields the optimal $q_{X|Y}(X \mid Y)$. Furthermore, the joint sufficient statistics are

$$
T(X,Y) = \left[f(X), g(Y), \mathrm{vec}(f(X)f(X)^T), \mathrm{vec}(f(X)g(Y)^T), \mathrm{vec}(g(Y)g(Y)^T)\right]
$$

and the sufficient statistics of a marginal distribution are

$$
T(X) = \left[f(X), \mathrm{vec}(f(X)f(X)^T)\right]
$$

which are simply the first and third sufficient statistic from the joint. Therefore moment matching the joint trivially moment matches the marginal, giving the optimal $q_X(X)$. □

**Lemma 3.2** (Flow Upper Bound). *Let $p(X,Y)$ be an arbitrary target distribution and $q_{X,Y}(X,Y)$ be the distribution of the form in Eqn. (14) with moment matched flow density $q_{Z,V}(Z,V)$. Then the bound in Lemma 2.1 is given by:*

$$
|I_{m+p} - I| \le \frac{1}{2}\log|2\pi e\Sigma_{zz}| + \frac{1}{2}\log\left|2\pi e\Sigma_{z|v}\right| - 2\mathbb{E}_{p_X}\left[\log|\nabla_x f(X)|\right] + C \tag{45}
$$

*Proof.* The upper bound in Lemma 2.1 is

$$
|I_{\mathrm{m}+p} - I| \le H_p(q_X(X)) + H_p(q_{X|Y}(q(X \mid Y)) + C
$$

This means we need access to the cross-entropy terms of the flow distribution

$$
H_p(q_X(X)) = -\int p(X)\log q_X(X)dx = -\int p(X)\log q_Z(f(X))|\nabla_x f(X)|\,dx \tag{46}
$$

$$
= H_p(q_Z(Z)) - \mathbb{E}_p\left[\log|\nabla_x f(X)|\right] = H_{q_z}(q_Z(Z)) - \mathbb{E}_p\left[\log|\nabla_x f(X)|\right] \tag{47}
$$

$$
= \frac{1}{2}\log|2\pi e\Sigma_{zz}| - \mathbb{E}_p\left[\log|\nabla_x f(X)|\right] \tag{48}
$$

$$
H_p(q_{X|Y}(X \mid Y)) = -\int p(X,Y)\log q_{X|Y}(X \mid Y)dx \tag{49}
$$

$$
= -\int p(X,Y)\log q_Z(f(X) \mid g(Y))|\nabla_x f(X)|\,dx \tag{50}
$$

$$
= H_p(q_{Z|V}(Z \mid V)) - \mathbb{E}_p\left[\log|\nabla_x f(X)|\right] \tag{51}
$$

$$
= H_{q_{Z|V}}(q_{Z|V}(Z \mid V)) - \mathbb{E}_p\left[\log|\nabla_x f(X)|\right] \tag{52}
$$

$$
= \frac{1}{2}\log\left|2\pi e\Sigma_{z|v}\right| - \mathbb{E}_p\left[\log|\nabla_x f(X)|\right] \tag{53}
$$

Here, we recognize that $H_p(q_{Z|V}(Z \mid V))$ is a moment matched Gaussian, so Lemma A.1 applies to the cross-entropy. With the cross entropy terms, we can now plug into the upper bound

$$
|I_{\mathrm{m}+p} - I| \le H_p(q_X(X)) + H_p(q_{X|Y}(q(X \mid Y)) + C \tag{54}
$$

$$
= \frac{1}{2}\log|2\pi e\Sigma_{zz}| + \frac{1}{2}\log\left|2\pi e\Sigma_{z|v}\right| - 2\mathbb{E}_p\left[\log|\nabla_x f(X)|\right] + C \tag{55}
$$

□

The equations for the cross entropies, Eqn. (48) and Eqn. (53), allow for us to explicity write the form of $I_{\text{JVF}}$

$$I_{\text{JVF}}(X,Y) = H_p(q_X(X)) - H_p(q(X \mid Y)) \tag{56}$$

$$= \frac{1}{2}\log|2\pi e\Sigma_{zz}| - \mathbb{E}_p\left[\log|\nabla_x f(X)|\right] - \frac{1}{2}\log\left|2\pi e\Sigma_{z|v}\right| + \mathbb{E}_p\left[\log|\nabla_x f(X)|\right] \tag{57}$$

$$= \frac{1}{2}\log|2\pi e\Sigma_{zz}| - \frac{1}{2}\log\left|2\pi e\Sigma_{z|v}\right| \tag{58}$$

We see that this is simply the MI of the Gaussian $q_{Z,V}(Z,V)$ which is the propety of MI invariance under invertible transformations (Czyż et al., 2023).

**Lemma 3.3.** *Let $p(X,Y)$ be any model and $q_{X,Y}(X,Y)$ of the form in Eqn. (14) with base distribution $q_{Z,V}(Z,V)$, then the following bound holds:*

$$|I_{JVF} - I| \leq 2H_p(q_{Z,V}(Z,V)) - 2\mathbb{E}_{p_X}\left[\log|\nabla_x f(X)|\right] - 2\mathbb{E}_{p_Y}\left[\log|\nabla_y g(Y)|\right] + C \tag{59}$$

*For a Gaussian base $q_{Z,V}(Z,V) = \mathcal{N}(\mu, \Sigma)$ the tightest bound of the form Eqn. (18) is given by the moment-matched flow distribution with $\Sigma = \text{Cov}_p(f(X), g(Y))$ and takes the form:*

$$|I_{JVF} - I| \leq \log|2\pi e\Sigma| - 2\mathbb{E}_{p_X}\left[\log|\nabla_x f(X)|\right] - 2\mathbb{E}_{p_Y}\left[\log|\nabla_y g(Y)|\right] + C \tag{60}$$

*where $C = 2H_p(p(X,Y))$. This bound is tight when $q_{X,Y}(X,Y) = p(X,Y)$ almost surely.*

*Proof.* Since JVF assumes and parameterizes a joint distribution $q_{X,Y}(X,Y)$, we have access to the symmetric definition of MI

$$I_{\text{JVF}}(X,Y) = H_p(q_X(X)) - H_p(q_{X|Y}(X \mid Y)) \tag{61}$$

$$= H_p(q_X(X)) - H_p(q_{X,Y}(X,Y)) + H_p(q_Y(Y)) \tag{62}$$

$$= H_p(q_Y(Y)) - H_p(q_{X,Y}(y \mid x)) = I_{\text{JVF}}(Y,X) \tag{63}$$

For this proof, we simply use $I_{\text{JVF}}(X,Y)$ and $I_{\text{JVF}}(Y,X)$ to differentiate between the two symmetric forms of JVF but do not necessarily hold this notation constant elsewhere. So we will apply Lemma 2.1 twice

$$|I_{\text{JVF}}(X,Y) - I(X,Y)| \leq 2|I_{\text{JVF}}(X,Y) - I(X,Y)| \tag{64}$$

$$= |I_{\text{JVF}}(X,Y) - I(X,Y)| + |I_{\text{JVF}}(Y,X) - I(Y,X)| \tag{65}$$

$$\leq H_p(q_X(X)) + H_p(q_{X|Y}(X \mid Y)) + C_1 \tag{66}$$

$$+ H_p(q_Y(Y)) + H_p(q_{Y|X}(y \mid x)) + C_2 \tag{67}$$

$$= 2H_p(q_{X,Y}(X,Y)) + C \tag{68}$$

where $C = C_1 + C_2$, $C_1 = -H_p(p(X)) - H_p(p(X \mid Y))$, and $C_2 = -H_p(p(Y)) - H_p(p(y \mid x))$ which come directly from bound applied to each term. We also were able to combine $H_p(q_X(X))$ and $H_p(q_{Y|X}(y \mid x))$ into $H_p(p_{X,Y}(X,Y))$ and likewise for $H_p(q_Y(Y))$ and $H_p(q_{X|Y}(X \mid Y))$. Since this was two applications of Lemma 2.1, we know that moment matching is optimal for bound and tight when $q_X(X) = p(X)$, $q_Y(Y) = p(Y)$, $q_{X|Y}(X \mid Y) = p(X \mid Y)$, and $q_{Y|X}(y \mid x) = p(y \mid x)$. Finally, deriving the explicit form of the bound, we get

$$|I_{\text{JVF}}(X,Y) - I(X,Y)| \leq 2H_p(q_{X,Y}(X,Y)) + C \tag{69}$$

$$= 2\mathbb{E}_p\left[-\log q_{X,Y}(X,Y)\right] + C \tag{70}$$

$$= 2\mathbb{E}_p\left[-\log q_{Z,V}(f(X), g(Y))|\nabla_x f(X)||\nabla_y g(Y)|\right] + C \tag{71}$$

$$= 2H_p(q_{Z,V}(Z,V)) - 2\mathbb{E}_p\left[\log|\nabla_x f(X)||\nabla_y g(Y)|\right] + C \tag{72}$$

$$= \log|2\pi e\Sigma| - 2\mathbb{E}_p\left[\log|\nabla_x f(X)|\right] - 2\mathbb{E}_p\left[\log|\nabla_y g(Y)|\right] + C \tag{73}$$

Where again, we used Lemma A.1 since $q_{Z,V}(Z,V)$ is a moment matched Gaussian. $\square$

### A.4 EXTERNAL SUPPORTING PROOFS

To prove the optimality of moment matching for our estimators in Theorem 2.2 and Theorem 3.1, we relied on the generalized results from Dahlke et al. (2023). We include related results from that paper in this section for convenience of understanding the proofs of Theorem 2.2 and Theorem 3.1.

**Theorem A.2.** *Let $q_m(X)$ be in the exponential family with statistics $T(X)$, then for any $p(X)$, the optimal $I_{marg}^*$ is given by moment matching:*

$$\mathbb{E}_{q_m(X)}[T(X)] = \mathbb{E}_{p(X)}[T(X)]$$

*Proof.* Since $H_p(X)$ is constant in $q_m$ we have,

$$\underset{q_m}{\operatorname{argmin}} H_p(q_m(X)) = \underset{q_m}{\operatorname{argmin}} H_p(q_m(X)) - H_p(X) = \underset{q_m}{\operatorname{argmin}} \mathrm{KL}(p(X) \| q(X)) \tag{74}$$

It is a known result, as proven in (Bishop & Nasrabadi, 2006), that for exponential families, $\mathbb{E}_{q_m(X)}[T(X)] = \mathbb{E}_{p(X)}[T(X)]$ minimizes $\mathrm{KL}(p(X) \| q(X))$. $\qquad\square$

**Lemma A.3.** *If $q_p(X \mid Y)$ takes the form of*

$$q_{X|Y}(X \mid Y) = q_{X|Y}(X \mid Y; \eta) = \frac{q_{X,Y}(X, Y; \eta)}{q_Y(Y; \eta)} \tag{75}$$

*where $q_{X,Y}(X, Y; \eta)$ is in the exponential family, then the minimization of $H_p(q_p(X|Y))$ occurs when*

$$\mathbb{E}_{p(Y)}\left[\mathbb{E}_{q_p(X|Y)}[T(X, Y)]\right] = \mathbb{E}_{p(X,Y)}[T(X, Y)] \tag{76}$$

*Proof.* The goal is to minimize $H_p(q_p(X|Y))$ where $q_p(X|Y)$ is generated from $q(X, Y; \eta)$ in the exponential family. We will find the minimizing parameters of this distributions. We appeal to the property of exponential families that $\frac{\partial}{\partial \eta} A(\eta) = \mathbb{E}_{q(X,Y)}[T(X, Y)]$

$$\frac{\partial}{\partial \eta}(H_p(q_p(X|Y))) = -\frac{\partial}{\partial \eta} \int p(X, Y) \log(q_p(X|Y)) = -\int p(X, Y) \frac{\partial}{\partial \eta} \log\left(\frac{q(X, Y; \eta)}{q(Y; \eta)}\right) \tag{77}$$

$$= -\int p(X, Y) \frac{\partial}{\partial \eta}\left(\log(h(X, Y)) + \eta^T T(X, Y) - A(\eta) - \log(q(Y; \eta))\right) dxdy \tag{78}$$

$$= -\int p(X, Y)\left(T(X, Y) - \frac{\partial}{\partial \eta} A(\eta) - \frac{\partial}{\partial \eta} \log(q(Y; \eta))\right) dxdy \tag{79}$$

$$= -\mathbb{E}_{p(X,Y)}[T(X, Y)] + \mathbb{E}_{q(X,Y)}[T(X, Y)] +$$
$$\int p(X, Y) \frac{1}{q(Y; \eta)} \frac{\partial}{\partial \eta}\left(\int h(X', Y) \exp\left(\eta^T T(X', Y) - A(\eta)\right) dx'\right) dxdy \tag{80}$$

$$= -\mathbb{E}_{p(X,Y)}[T(X, Y)] + \mathbb{E}_{q(X,Y)}[T(X, Y)] +$$
$$\int p(X, Y) \frac{1}{q(Y; \eta)}\left(\int q(X', Y; \eta)\left(T(X', Y) - \frac{\partial}{\partial \eta} A(\eta)\right) dx'\right) dxdy \tag{81}$$

$$= -\mathbb{E}_{p(X,Y)}[T(X, Y)] + \mathbb{E}_{q(X,Y)}[T(X, Y)] +$$
$$\int p(X, Y)\left(\int q(X'|Y)\left(T(X', Y) - \mathbb{E}_{q(X,Y)}[T(X, Y)] dx'\right) dxdy\right) \tag{82}$$

$$= -\mathbb{E}_{p(X,Y)}[T(X, Y)] + \mathbb{E}_{p(Y)}\left[\mathbb{E}_{q_p(X|Y)}[T(X, Y)]\right] \tag{83}$$

The zero derivative yields $\mathbb{E}_{p(X,Y)}[T(X, Y)] = \mathbb{E}_{p(Y)}\left[\mathbb{E}_{q_p(X|Y)}[T(X, Y)]\right]$ which is the stationary condition. It now remains to show that the objective is convex in $\eta$. Expanding the form of $H_p(q(X \mid Y))$ we have the objective,

$$\min_{\eta} -\mathbb{E}_p\left[\log(h(X, Y)) + \eta^T T(X, Y) - A(\eta) - \log(q(Y; \eta))\right] \tag{84}$$

The term $\eta^T T(X, Y)$ is linear in $\eta$. Convexity of $A(\eta)$ in $\eta$ is a standard property of the exponential family, however we will show a constructive proof that $A(\eta) + \log(q(Y; \eta))$ is convex using Hölder's inequality. Let $\eta = \lambda \eta_1 + (1 - \lambda)\eta_2$ where $\lambda \in [0, 1]$ and $\eta_1, \eta_2$ in the convex set of valid exponential

family parameters of $q$ then:

$$A(\eta) + \log(q(Y;\eta)) = A(\eta) + \log\left(\int h(X,Y)\exp(\eta^T T(X,Y) - A(\eta))\,dx\right) \tag{85}$$

$$= A(\eta) + \log\left(\exp(-A(\eta))\int h(X,Y)\exp(\eta^T T(X,Y))\,dx\right) \tag{86}$$

$$= \log\left(\int h(X,Y)\exp(\eta^T T(X,Y))\,dx\right) \tag{87}$$

$$= \log\left(\int (h(X,Y)\exp(\eta_1^T T(X,Y)))^\lambda (h(X,Y)\exp(\eta_2^T T(X,Y)))^{(1-\lambda)}\,dx\right) \tag{88}$$

$$\le \lambda\log\left(\int h(X,Y)\exp(\eta_1^T T(X,Y))\,dx\right) + (1-\lambda)\log\left(\int h(X,Y)\exp(\eta_2^T T(X,Y))\,dx\right) \tag{89}$$

$$= \lambda(A(\eta_1) + \log q(Y;\eta_1)) + (1-\lambda)(A(\eta_2) + \log q(Y;\eta_2)) \tag{90}$$

Thus convexity holds in $\eta$ and the stationary conditions are globally optimal.

$$\mathbb{E}_{p(X,Y)}[T(X,Y)] = \mathbb{E}_{p(Y)}\left[\mathbb{E}_{q_p(X|Y)}[T(X,Y)]\right] \tag{91}$$

$\square$

**Theorem A.4.** *Let $q(X,Y)$ be in the exponential family with sufficient statistics, $T(X,Y) = [\tau(X),\tau(Y),\tau(X,Y)]^T$ where $\tau(X)$ are the sufficient statistics dependent only on $x$, $\tau(Y)$ only on $y$, and $\tau(X,Y)$ on both. Further, let the posterior expected statistics be a linear combination of marginal statistics as in,*

$$\mathbb{E}_{q_p(X|Y)}[T(X,Y)] = \sum_i^k g_i(\eta)\tau_i(Y) \tag{92}$$

*where $\tau_i(Y)$ is the $i^{th}$ component of $\tau(Y)$ and $g_i(\eta)$ are functions of only the parameter $\eta$. Then, the optimal variational distribution, $q_p$, for $I_{post}$ is defined by joint moment matching: $\mathbb{E}_{p(X,Y)}[T(X,Y)] = \mathbb{E}_{q(X,Y)}[T(X,Y)]$.*

*Proof.* From Lemma A.3, we know that $\mathbb{E}_{p(X,Y)}[T(X,Y)] = \mathbb{E}_{p(Y)}\left[\mathbb{E}_{q_p(X|Y)}[T(X,Y)]\right]$ is the optimality condition. Let us now show that the condition in Eqn. (92) implies that joint moment matching satisfies the optimality condition of Eqn. (76)

$$\mathbb{E}_{p(X,Y)}[T(X,Y)] = \mathbb{E}_{q(X,Y)}[T(X,Y)] \tag{93}$$

$$= E_{q(Y)}\left[\mathbb{E}_{q_p(X|Y)}[T(X,Y)]\right] = \mathbb{E}_{q(Y)}\left[\sum_i^k g_i(\eta)\tau_i(Y)\right] \tag{94}$$

$$= \sum_i^k g_i(\eta)\mathbb{E}_{q(Y)}[\tau_i(Y)] = \sum_i^k g_i(\eta)\mathbb{E}_{p(Y)}[\tau_i(Y)] \tag{95}$$

$$= \mathbb{E}_{p(Y)}\left[\sum_i^k g_i(\eta)T_i(Y)\right] = \mathbb{E}_{p(Y)}\left[\mathbb{E}_{q_p(X|Y)}[T(X,Y)]\right] \tag{96}$$

So with Lemma A.3 and the assumption of the posterior expected statistics being a linear combination of joint statistics (Eqn. (92)) results in $\mathbb{E}_{p(X,Y)}[T(X,Y)] = \mathbb{E}_{q(X,Y)}[T(X,Y)]$ being the optimal conditions. $\square$

**Theorem A.5.** *Let $q_{\mathrm{marg}}(X)$ and $q(X,Y)$ be exponential family distributions where the joint sufficient statistics are $T(X,Y) = [\tau(X),\tau(Y),\tau(X,Y)]^T$ and $\tau(X)$ are the sufficient statistics dependent only on $x$, $\tau(Y)$ only on $y$, and $\tau(X,Y)$ on both. Further, let $q(X,Y)$ satisfy the linear conditional expectations property*

$$\mathbb{E}_{q_{\mathrm{post}}(X|Y)}[T(X,Y)] = \sum_i^k g_i(\eta)\tau_i(Y) \tag{97}$$

*where $\eta$ are the natural parameters of $q(X,Y)$ and $g_i(\eta)$ are arbitrary functions dependent only on the natural parameters. Then, moment matching the joint $q(X,Y)$ and marginal $q_m(X)$*

$$\mathbb{E}_{p(X,Y)}[T(X,Y)] = \mathbb{E}_{q(X,Y)}[T(X,Y)]$$

$$\mathbb{E}_{p(X)}[T(X)] = \mathbb{E}_{q_{\mathrm{marg}}(X)}[T(X)]$$

*yield optimal* $q_{\mathrm{post}}(X \mid Y) \propto q(X, Y)$ *and* $q_{\mathrm{marg}}(X)$ *that minimize the bound on* $I_{m+p}$ *in Lemma 2.1.*

*Proof.* We break this down into the two cases of Theorem A.2 and Theorem A.4. Notice that the variational distributions $q_m(X)$ and $q_p(X \mid Y)$ need not share a common joint $q(X, Y)$. So, let us use different natural parameters, $\eta_1$ and $\eta_2$, for each (i.e. $q_m(X) = q(X; \eta_1)$ and $q_p(X|Y) = q(X|Y; \eta_2)$). We optimize the bound in Lemma 2.1 with respect to both natural parameters, beginning with $\eta_1$:

$$\frac{\partial}{\partial \eta_1} \left( -\mathbb{E}_{p(X,Y)} \left[ \log q(X; \eta_1) + \log q(X \mid Y; \eta_2) \right] + C \right) = -\frac{\partial}{\partial \eta_1} E_{p(X,Y)} \left[ \log q(X; \eta_1) \right] \quad (98)$$

This is exactly the condition in Theorem A.2 which we know is solved by moment matching the marginal. Likewise, for $\eta_2$:

$$\frac{\partial}{\partial \eta_2} \left( -\mathbb{E}_{p(X,Y)} \left[ \log q(X; \eta_1) + \log q(X \mid Y; \eta_2) \right] + C \right) = -\frac{\partial}{\partial \eta_2} E_{p(X,Y)} \left[ \log q(X \mid Y; \eta_2) \right] \quad (99)$$

The above is the start of the proof for Lemma A.3 in Eqn. (77) and along with Eqn. (92) in Theorem A.4, we get that moment matching the joint finds the optimal $q_p$. Therefore, the optimization of Lemma 2.1 simply reduces to moment matching the marginal and the joint. $\square$

## B  EXPERIMENT DETAILS

All code can be found at `https://github.com/calebdahlke/FlowMI`.

### B.1  HIGH MUTUAL INFORMATION EXPERIMENT

We consider a synthetic test case where we can create a distribution with high mutual information. Consider a joint Gaussian where $X, Y \in \mathbb{R}^{15}$

$$p(X, Y) = \mathcal{N} \left( \begin{bmatrix} x \\ y \end{bmatrix} \middle| \mu := \begin{bmatrix} \mu_x \\ \mu_y \end{bmatrix}, \Sigma := \begin{bmatrix} \Sigma_{xx} & \Sigma_{xy} \\ \Sigma_{yx} & \Sigma_{yy} \end{bmatrix} \right) \quad (100)$$

where $\mu = 0$ is the zero vector, $\Sigma_{xx} = \Sigma_{yy} = I$, and $\Sigma_{xy} = \Sigma_{yx} = \rho I$. Then $I(X, Y) = -\frac{Dim_X}{2} ln(1 - \rho^2)$ so we are free to choose the value of MI by changing the dimension or by changing the correlation constant. We select $\rho = .95$ to achieve a MI of $I(X, Y) = 17.459$. We train all estimators using a batch size of $N = 256$ for a total of 3000 steps to ensure all estimators converged.

The critic based methods (DV, MINE, InfoNCE, and NWJ) all use a neural network with two hidden layers of the structure $[16, 8]$ with ReLU activation functions. Czyż et al. (2023) utilized a $lr = .1$ and batch size of $N = 256$ which we kept the same.

The normalizing flows utilized in JVF and NVF are Rational Quadratic Splines (Durkan et al., 2019) which learn 128 knots to parameterize the spline. The spline is bounded between $-8$ and $8$ where is is linear outside. The knots are learned from a neural network of size $[8, 8]$ with ReLU activation functions. To prevent overfitting of the flows, we utilize dropout with a rate of .2 and L2 Regularization with a factor of $10^{-5}$.

The neural parameters used in NVG and NVF are the mean and variance parameterized by two neural networks of size $[16, 8]$ with relu activation functions. The output of the neural network parameterizing the covariance is a lower triangular matrix to approximate the cholesky decomposition of $\Sigma$ where the diagonal elements have a Softplus activation applied to them to ensure the resulting matrix is semi-positive definite. These parameters did not have dropout performed but did have the same L2 Regularization with a factor of $10^{-5}$. NVG, JVF, and JVF all utilized a learning rate of .005 and reduced learning rate by a factor of .1 on test loss if no improvement was seen over 250 testing steps.

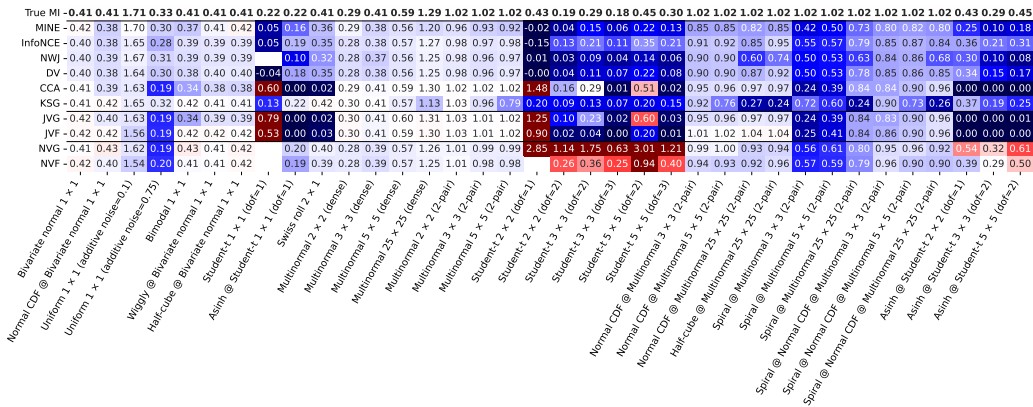

Figure 5: **MI Benchmark** runs against a selection of 36 different scenarios. Blue indicates an underestimate and red indicates on overestimate with the magnitude indicated by saturation. Blank cells indicate that the method produced divergent estimates.

## B.2 Mutual Information Benchmark

Czyż et al. (2023) created a large selection of MI benchmark tests with gound truth values via MI invariance properties under certain transformations. We will give a brief description of the experiments but in-depth information can be found in their paper.

**Bivariate Normal:** A simple $1 \times 1$ dimensional Gaussian with $\rho = .75$.

**Uniform Margins:** The Gaussian CDF, $\Psi(\cdot)$, is applied to Gaussian variables $X$ and $Y$, the resulting distributions are marginally uniform but not jointly uniform. Denoted as 'Normal CDF @ P'.

**Half-Cube Map:** The transformation $F(X) = |x|^{3/2}$ is applied to Gaussian variables with the goal of lengthening the tails. Denoted as 'Half-cube @ P'.

**Asinh Mapping:** To shorten tails, the inverse hyperbolic sine function $\operatorname{asinh}(X) = \log(X + \sqrt{1 + x^2})$ is applied. Denoted as 'Asinh @ P'.

**Wiggly Mapping:** To model non-uniform lengthscales, the mapping $F(X) = x + \sum a_i \sin(w_i x + \phi_i)$ is applied. Denoted as 'Wiggly @ P'

**Bimodal Variables:** In the inverse CDF of a two component GMM is applied to $X$ and $Y$ to create multimodality.

**Additive Noise:** The random variable $X \sim \operatorname{Uniform}(0, 1)$ is considerd along with noise $N \sim \operatorname{Uniform}(-\epsilon, \epsilon)$, then $Y = X + N$. Denoted 'Uniform (additive noise=$\epsilon$).

**Swiss Roll Embedding:** $X, Y \sim \operatorname{Uniform}(0, 1)$ and the Swiss roll embedding is performed on $X$.

**Multivariate Normal (2-pair):** $X$ and $Y$ are jointly Gaussian distributed with $\operatorname{Cor}(X_1, Y_1) = \operatorname{Cor}(X_1, Y_1) = .8$ and 0 correlation everywhere else.

**Multivariate Normal (Dense):** $X$ and $Y$ are jointly Gaussian distributed with all off diagonal correlations set to .5.

**Multivariate Student:** A multivariate Student-T distribution with degrees of freedom $\nu$.

**Spiral:** The transformation $F(X) = \exp(vA\|x\|^2)x$ where $A$ is a skew-symmetric matrix which 'mixes' the dimensions and $v$ is the rate at which the mixing occurs. Denoted 'Spiral @ P'.

The dimensions $X$ and $Y$ range from 1 to 25 to give a wide selection of applications. Each estimator is given 1000 samples per dimension with a train test split of 50-50. Each estimator is evaluated on the testing data every 250 gradients steps and early stopping is performed if the estimator test loss has not decreased within the previous 500 steps. All data is pre-processed by centering the data and scaling each dimension by it variance as performed in (Czyż et al., 2023). The same network structures are used as in the Large MI experiment.

## B.3 Location Finding

In the this experiment, we have 2 hidden objects in $\mathbb{R}^2$ which we wish to learn their locations $X = \{X_1, X_2\}$. Each source emits a signal that decays with respect to the inverse square law. From

and position, $d$, we can observe the superposition of all signals together

$$\mu(X, d) = b + \frac{1}{m + \|X_1 - d\|^2} + \frac{1}{m + \|X_2 - d\|^2} \tag{101}$$

where $b = .1$ is the background noise and $m = 10^{-4}$ controls the maximum signal. We can then normally observe the log intensity of the signal from the likelihood

$$\log y \mid X, d \sim \mathcal{N}(\log(\mu(X, d)), \sigma^2 \tag{102}$$

where $sigma = .5$ controls the noise in the observation. We condsider two settings for our experiments, *matched prior* and *mismatched prior*. In the case of the matched prior, every method assumes the prior $p(X) = \mathcal{N}(0, I)$ which corresponds to the true testing prior where testing $x$ are sampled from. In the mismatched prior setting, every method still assumes that $p(X) = \mathcal{N}(0, I)$ but the true testing $x$ are sampled from $\mathcal{N}(1, I)$. In practice, we often assume a convenient prior but that prior likely does not correspond well with the real world so an ideal BOED experiment should be able to adapt to this mismatch with minimal loss in information to still provide informative decision across a sequence of decisions. We notice that in this experiment JVG, NVG, and JVF perform significantly worse than NVF. To shed some light on this, we can look at the posterior prediction after making a single observation (Fig.4). The result of this observation increases the belief that sources are located in a radius away from the origin. Neither JVG nor NVG are flexible enough to capture this belief and instead produce Gaussian posteriors. Including flow JVF and NVF have increasesed flexibility to capture the non-Gaussian belief. We see that the additional degrees of freedom in NVF allows for a higher posterior belief in the vicinity of the true target locations, whereas JVF concentrates more away from the targets.

