# OpenReview forum: "Flow-based Variational Mutual Information: Fast and Flexible Approximations"
_ICLR.cc/2025/Conference — ICLR 2025 Poster_

### Official Review · Reviewer_9DeW · 2024-10-30

**Soundness:** 4
**Presentation:** 3
**Contribution:** 4
**Rating:** 8
**Confidence:** 5

**Summary:**

This paper focuses on developing a variational bound for the error between a model for mutual information and the true (unknown) mutual information. Through this bound the framework can be used to estimate the mutual information as well as to draw samples from the underlying distribution. A key element employed for achieving model flexibility in modeling the distribution (or samples therefrom) is the use of normalizing flows. This setup is coupled with multiple forms of moment matching, as applied to the approximated first and second order moments of the data, estimated from samples. Neural networks are employed in multiple ways: They are employed within the normalizing flows, and they (for one of the models) are used to model the mean and covariance within conditional distributions (employed within the NVF estimator). An interesting application of this technology is employed within adaptive Bayesian experimental design. Extensive and encouraging experimentation is shown.

**Strengths:**

The paper develops a sophisticated and well-grounded formulation for flexibly estimating the mutual information based on observed samples, and impressive experiments are performed.

**Weaknesses:**

Generally the paper is well written, but there are places it could be improved. Additionally, while the bibliography is comprehensive, there is some recent work that should be referenced and compared to (theoretically at least, while additional experiments are unnecessary).

**Questions:**

I enjoyed reading this paper. It was generally well done and well motivated. I have some suggestions for possible improvements.

1. For such a well thought through concept, I think aspects of the writing could be improved. My concern is where to use upper case X (or similar) for a random variable, and lower case x for an instantiation of that random variable. Just below Eq. 19, you write C=2H_p(p(X,Y)). And you use similar uppercase variables in Eq (18) for q_{Z,V}(Z,V). But, in many (most) other places you use lowercase variables, such as in (1), (2) and many other places you use H_p(p(x)) and h_p(p(x|y)).

At minimum you need to be consistent, and this use of notation is not consistent. I think the notation in (18) and below Eq (19) is better.

If we take Cover and Thomas, "Elements of Information Theory," (2nd Edition),  as a guide -- as this is a "bible" of information theory:

Mutual information is I(X;Y) : importantly, (X,Y) are uppercase, because these are RVs, not instantiations of RVs.
Entropy is H(X): Importantly, X is upper case.

I know that sometimes you may wish to emphasize what distribution you use for these measures, so (like under Eq 19) I suggest you use H(p(X,Y)), but most importantly: use uppercase (X,Y)

As an example, your Eq 1 should (in my opinion) be H_p(p(X))-H_p(p(X|Y)). This notation is consistent with Cover & Thomas.

In (2), you do have I(X,Y). But you have H_p(q_{marg}(x)), for example. I think this should be H_q(q_{marg}(X))

This issue with notation is everywhere in the paper. Usually it is not consistent with Cover & Thomas, and sometimes (rarely) it is.

In my view, this is a very nice paper. However, the above is very distracting for readers who have a background in information theory (largely your readership). I think you want to write this such that it is consistent with the information theory literature. I suggest you carefully go through and revise. As examples:

q_{marg}(x) -> q_{marg}(X)
H_p(p(x|y)) -> H_p(p(X|Y))
q(x,y) -> q(X,Y)

etc etc. Above are just a tiny subset.

2. Your bibliography is extensive. However, you missed

Pengyu Cheng, Weituo Hao, Shuyang Dai, Jiachang Liu, Zhe Gan, Lawrence Carin, "CLUB: A Contrastive Log-ratio Upper Bound of Mutual Information," ICML 2020

That paper also uses sophisticated NNs and addresses a similar problem. Please discuss how your approach relates and compares to this approach.

3. In the Abstract, you write "In this paper, we introduce two new classes of variational estimators
that extend the previous Gaussian-based variational estimators with Normalizing
Flows." This is ambiguous -- did the previous Gaussian-based methods use normalizing flows? No, your method uses normalizing flows. Please consider changing this to: "In this paper, we introduce new classes of variational estimators based on
Normalizing Flows that extend the previous Gaussian-based variational estimators."

4. Eq (5) is basically the same idea as employed in Gaussian processes to model the distribution of x based on observed y. Maybe you should note this (minor). It could help the reader, particularly GP experts, understand this.

---

> ### Author Response · Authors · 2024-11-19
>
> **Notation and Abstract Changes**
>
> We thank the reviewer for their careful review and detailed suggestions. We agree with the notation inconsistencies and have made the recommended changes to improve clarity. Specifically, we have updated the notation from $q_{marg}(x) \rightarrow q_{marg}(X)$ and $H_p(p(x|y)) \rightarrow  H_p(p(X|Y))$ among other adjustments. Additionally, we have revised the abstract to eliminate any ambiguity, as per the reviewer’s suggestions.
>
> **Relation to CLUB**
>
> We included the suggested paper in our bibliography and have added a short discussion of the method in our Related Work (Section 5). CLUB proposes an upper bound of MI
>
> $$I(x,y)\leq I_{CLUB}(x,y) = \mathbb{E}\_{p(x,y)} [\log p(y\mid x)]-\mathbb{E}\_{p(x)p(y)} [\log p(y\mid x)] $$
>
> This upper bound has two notable drawbacks: it is tight iff $x$ and $y$ are independent and requires an explicit $p(y\mid x)$. CLUB circumnavigates the second issue by using a variational distribution, similar to NVG, with $\mu$ and $\Sigma$ parameterized by neural networks. One could potentially extend their variational distribution with our NVF distribution, however this approximation still only converges to the true MI when $x$ and $y$ are independent, regardless of how accurate the variational distribution is.

---

### Official Review · Reviewer_uioR · 2024-11-04

**Soundness:** 3
**Presentation:** 3
**Contribution:** 2
**Rating:** 5
**Confidence:** 3

**Summary:**

The paper introduces two classes of variational estimators for mutual information that utilize normalizing flows to extend Gaussian-based methods. These estimators are shown to maintain guarantees while improving the flexibility of the variational distribution. Experimental validations demonstrate the efficacy of these methods on mutual information (MI) problems and in applications like Bayesian Optimal Experimental Design (BOED).

**Strengths:**

Theoretical Innovation: The introduction of flow-based variational estimators represents a theoretical enhancement over traditional Gaussian-based methods. The authors effectively integrate normalizing flows, which are known for their flexibility in modeling complex distributions.

Comprehensive Validation: The paper presents a robust set of experiments that not only compare the proposed methods against existing estimators but also demonstrate their superiority in scenarios where others fail, particularly in large MI estimation problems.

**Weaknesses:**

Assumptions Limitation: The estimators assume the availability of a joint distribution, which might not be practical or accurate in all real-world scenarios. Although one of the proposed estimators, NVF, relaxes this assumption, the general applicability of the method under different conditions remains underexplored.

Robustness Concerns: The discussion on the robustness of the models, particularly under model misspecifications and with non-Gaussian noise (e.g., Student-T distributions), is not thoroughly addressed. The capability of handling Non-Gaussian (especially heavy-tailed distributions) is not comprehensively validated.

**Questions:**

How do the proposed estimators perform under different levels of model misspecification or with datasets characterized by heavy-tailed noise? Are there particular settings or types of data where the performance of these estimators might degrade significantly? For example, 4th moment bounded heavy tails or 2nd moment bounded heavy tails.

---

> ### Author Response · Authors · 2024-11-19
>
> **Estimators assume the availability of a joint distribution**
>
> We thank the reviewer for their time and helpful insights. The reviewer correctly notes that only JVF assumes access to a consistent joint distribution, but NVF does not.  If a joint distribution is required then the practitioner is free to use JVF.  If a joint distribution is not required then NVF is a feasible option.
>
> **Performance under Heavy-Tailed Distributions**
>
> Our Mutual Information Benchmark experiment (Section 6.2) includes an extended test set in the appendix (Figure 5) that considers various Student-t distributions with different degrees of freedom. We observe that all methods perform poorly under unnormalized data with heavy-tailed distributions. To address this known limitation Czyż et al. (2023) proposed alternative experiments called 'Asinh @ Student-T' which normalizes the data to remove fat tails. In these experiments, despite model misspecification, our NVF estimator provides substantially improved estimates, while other estimators fail. This suggests that NVF is capable of accurately estimating MI even in the presence of model mismatch, provided the data is appropriately normalized. We are happy to include additional discussion to this effect in the main text if the reviewer believes this would offer further clarity.

---

> > ### Author Response · Authors · 2024-12-03
> > **Follow-up**
> >
> > Dear Reviewer uioR,
> >
> > We thank you again for taking the time to review our work, and provide valuable insights. We are hopeful that our rebuttal addresses the concerns you raised in your initial review.
> >
> > As the discussion period draws to a close we kindly ask that you please consider whether our responses have addressed your concerns.
> >
> > If your concerns have been addressed we would be truly grateful if you would reconsider your initial evaluation of the paper based on the clarification provided in our responses.
> >
> > Thanks again! Authors

---

### Official Review · Reviewer_CMGX · 2024-11-04

**Soundness:** 2
**Presentation:** 3
**Contribution:** 3
**Rating:** 6
**Confidence:** 3

**Summary:**

This paper introduces a new approach to estimate Mutual Information (MI) using flow-based variational methods. Existing methods like variational and critic-based approaches use restrictive Gaussian approximations and struggle with large MI values.
The paper proposes two main methods:
Joint Variational Flow (JVF): Extends the Joint Variational Gaussian (JVG) estimator by using normalising flows while maintaining efficient moment-matching updates
Neural Variational Flow (NVF): Builds on the Neural Variational Gaussian (NVG) estimator to provide a more expressive distribution.

**Strengths:**

Originality:
The paper creatively combines normalising flows with existing variational MI estimators (JVG and NVG).
The approach to stabilising training through the modified bound in Lemma 3.3 is interesting

Significance:
The approach shows robustness to model mismatch which can be important for real-world applications.
Contributions can extend beyond MI estimation to general flow-based density estimation

Quality:
The paper provides thorough theoretical development with clear proofs and lemmas.
Honest discussion of limitations and failure cases

Clarity:
Experiments section has a clear motivation for each of the toy test

**Weaknesses:**

The experiments although quite interesting, fail to provide sufficient evidence for practical benefits of the proposed approach. The argument for practical benefits hinges on the BOED application. Unfortunately the paper fails to provide convincing experimental evidence for the BOED application. More specifically, the only test is on a location-finding scenario while previous works like Foster et al. (2021) and Ivanova et al. (2021) evaluate on multiple BOED benchmarks:

The presentation of the experiments is a bit sloppy as there are no error bars or ablation studies on key parameters such as the dimensionality of the problem and the impact of number of sequential decisions

In my view these type of experiments are more suited for AISTATS than ICLR. For ICLR, I would encourage the authors to provide more realistic benchmarks similar to those used in Foster et al. (2021) and Ivanova et al. (2021).

**Questions:**

I would be happy to increase my score if the authors can improve the experiments section by 1.) Testing on at least one more benchmark setting used in the prior works. 2.) Add ablations focusing on the impact of number of sequential decisions (with error bars please).

---

> ### Author Response · Authors · 2024-11-19
>
> **Multiple BOED benchmarks**
>
> We thank the reviewer for their time and useful feedback. Foster et al. (2021) and Ivanova et al. (2021) focus on sequential decision-making experiments, reflecting the core contributions of their work, which are centered around training policy-based methods. The core contributions of our paper are MI estimators, and we believe the experiments should reflect this position. Our focus is on a thorough analysis of MI estimation (Sections 6.1 and 6.2), then further supported with an application to BOED in Section 6.3.
>
> **These type of experiments are more suited for AISTATS than ICLR**
>
> We believe that the experiments are appropriate for ICLR.  In particular our insight that policy based approaches are sensitive to mismatched priors in the sequential decision making setting (Sec. 6.3) will benefit from the wider audience of ICLR. We believe that highlighting this shortcoming of policy based approaches and how our direct myopic approach is more stable is a beneficial insight to the problem at hand.
>
> **The presentation of the experiments is a bit sloppy as there are no error bars**
>
> We have modified Figure 2 for our Large MI experiment (Section 6.1) to include confidence intervals. For the MI benchmark test (Section 6.2), we comment in line 423 "(due to space we omit standard errors)". This mirrors how the benchmark tests were presented in Czyż et al. (2023). Finally, we do report standard errors in our BOED experiment (Section 6.3) as presented in Table 2.

---

> > ### Comment · Reviewer_CMGX · 2024-11-27
> > **Response**
> >
> > Thanks for you response, I am happy to increase my score. But I still think multiple BOED benchmarks will greatly help the significance of your contributions.

---

### Official Review · Reviewer_Gj86 · 2024-11-04

**Soundness:** 2
**Presentation:** 2
**Contribution:** 2
**Rating:** 5
**Confidence:** 3

**Summary:**

This paper introduces two new classes of variational estimators, Joint Variational Flow (JVF) and Neural Variational Flow (NVF) via combining normalizing flows with moment-matching based joint Gaussian estimators. The inclusion of flow has significantly increased the flexibility of the estimator, enabling more accurate approximation for complex interactions. And moment matching strategy allows efficient computation. The author(s) compared the proposed JVF and NVF with popular MI estimator on multiple synthetic datasets and discussed their pros and cons.

**Strengths:**

* The introduction of normalizing flow greatly enhanced the flexibility of Gaussian variational MI estimator
* To mitigate the stability issues observed for the original JVF estimator, author(s) proposed an alternative formulation where the Jacobian of both flows appear in the objective
* The proposed estimators also give distribution approximation of p(x,y), where people can easily draw samples (for prediction or other simulation-based tasks).
* Adequate coverage of relevant literature

**Weaknesses:**

* [Minor] Classification of MI estimators. In the introduction, the author(s) discussed critic-based MI-estimators and variational MI-estimators. However, many of those “critic-based” estimators (e.g., DV, NWJ, InfoNCE, etc.) actually belong to the variational family, see [B. Poole 2019]. These “variational” estimators described by the author(s) should be considered as parametric estimations.
* [Major] Practical value of the proposed estimators. In practical applications, people often seek to optimize the MI, instead of just estimating its value. For example, one major application of MI optimization is representation learning, where the MI between two related data views are optimized — it is the learned representation that matters, not the estimated MI. “Critic-based” MI estimators are heavily used in such scenarios, given their simple formulation and strong performance for real world complex data. To the best of my knowledge, normalizing flow based approaches rarely moved beyond academic settings. This paper also did not provide any empirical/theoretical evidence that proposed flow-based estimator can deliver competitive performance with popular alternatives such as InfoNCE in representation learning scenario — given ICLR is the International Conference on Learning Representations, the author(s) have to check this box.
* [Major] Experiments. The experiment section needs to be strengthened. Currently all the experiments are synthetic data, which is inadequate to demonstrate the robustness of the proposed methods to complex real world data. Also while the author(s) mentioned two variants of JVF, there is not empirical comparison between the two. For the location finding experiment, the author(s) should show the comparison posterior distribution of source points after all ten observations.

**Questions:**

Can the author(s) clarify how their proposed methods can handle scenarios where the data distribution resides on a lower-dimensional manifold?

---

> ### Author Response · Authors · 2024-11-19
>
> **Classification of MI estimators**
>
> We thank the reviewer for their thoughtful comments. In the reference provided by the reviewer (Poole et al., 2019), all the estimators mentioned (DV, NWJ, and NCE) are referred to as critic-based methods, which is why we have adopted this naming convention. We agree that the use of "variational" terminology may be confusing.  We have revised the manuscript to adopt the naming convention of Song and Ermon (2020). We use the term "discriminative" to refer to estimators that approximate the log-density ratio such as DV, NWJ, and NCE.  We use the term "generative" to refer to methods that estimate the generative densities, such as our proposed estimators.
>
> **Competitive performance with popular alternatives such as InfoNCE**
>
> The goal of our High Mutual Information Experiment (Section 6.1) is to highlight the settings where discriminative methods, like InfoNCE, fail. It is true that in some applications, such as BOED, the  MI value is less useful than the relative value across different decisions.  However, the discriminative estimators under consideration also do not guarantee stable MI rankings. By ensuring accurate MI estimation we also ensure that our proposed methods provide more stable MI rankings.
>
> **Two variants of JVF**
>
> We believe the reviewer is referring to the two upper bounds introduced for training JVF in Lemma 3.2 and Lemma 3.3. Lemma 3.2 is a direct extension of the upper bound in Lemma 2.1. We clarify in the paper (line 218) that "the bound in Lemma 3.2 only considers the log determinant of f(x), and in practice, we found that g(y) tends to overfit during learning."  Due to this overfitting issue we observe that the bound in Lemma 3.2 yields poor results.  Thus, we do not recommend the bound in Lemma 3.2 and so we do not include it in the experimental results.  We will make this more clear in a final version of the paper.
>
> **Data distributions on a lower-dimensional manifolds**
>
> We address this concern in our Mutual Information Benchmark experiment (Section 6.2), where we consider an example distribution with embedded data. Specifically, we use the "Swiss roll" distribution, which embeds 1-dimensional data into a 2-dimensional space. Our JVF estimator struggles to accurately capture this embedding. However, the NVF estimator is more capable of handling such embeddings and can provide more accurate estimations across dimensions. We would like to emphasize that, although we used Neural Spline Flows (Durkan et al., 2019) in this paper, our results do not depend on any specific flow structure. To improve JVF’s handling of data embeddings, we suggest considering the framework introduced in "Normalizing Flows Across Dimensions" by Cunningham et al. (ICML 2020).
>
> **Relevance to ICLR**
>
> We believe the work to be well-suited to ICLR as our estimators rely on density estimation, a form of learning representation in its own right.  Moreover, the focus of ICLR has broadened beyond just "learning representations" as the conference has grown.  This shift is evidenced by the subject areas in the call for papers, which includes the subject area of our work "probabilistic methods (Bayesian methods, variational inference, sampling, UQ, etc.)".  We note that MI estimation is a form of uncertainty quantification (UQ) and our methods are decidedly Bayesian and variational.  Finally, we note that we provide a reference (Song and Ermon, 2020), which is well-cited and published at ICLR, also addresses the problem of MI estimation using variational methods.

---

> > ### Author Response · Authors · 2024-12-03
> > **Followup**
> >
> > Dear Reviewer Gj86,
> >
> > We thank you again for taking the time to review our work, and provide valuable insights. We are hopeful that our rebuttal addresses the concerns you raised in your initial review.
> >
> > As the discussion period draws to a close we kindly ask that you please consider whether our responses have addressed your concerns.
> >
> > If your concerns have been addressed we would be truly grateful if you would reconsider your initial evaluation of the paper based on the clarification provided in our responses.
> >
> > Thanks again! Authors

---

### Author Response · Authors · 2024-11-25
**Thank you!**

We thank all reviewers for their valuable time and useful feedback.  As the discussion period draws to a close please review our responses and manuscript revisions and consider whether your concerns have been addressed.  Thanks again!

---

### Meta-Review · Area_Chair_ryBH · 2024-12-25

**Metareview:**

This paper presents a new mutual information (MI) estimator. Varitional estimator of  MI use an approximate distribution to create various bounds of MI. While classical methods use Gaussian distributions for this purpose, authors propose using normalizing flows to obtain these bounds. Authors introduce Joint Variational Flow (JVF) and Neural Variational Flow (NVF)  by combining normalizing flows with moment-matching based joint Gaussian estimators.

**Additional Comments On Reviewer Discussion:**

Reviewers agreed that the paper has a good contribution with Reviewer 9DeW championing the paper. Reviewer CMGX increased their score based on the rebuttal.

---

### Decision · Program_Chairs · 2025-01-22

Accept (Poster)